# miRNA profile is altered in a modified EAE mouse model of multiple sclerosis featuring cortical lesions

**Nicola S Orefice[1†§\*], Owein Guillemot-Legris[2†], Rosanna Capasso[3], Pauline Bottemanne[2], Philippe Hantraye[1], Michele Caraglia[3], Giuseppe Orefice[4], Mireille Alhouayek[2‡], Giulio G Muccioli[2‡]**

[1]CEA, Fundamental Research Division (DRF), Institute of Biology Francois Jacob, Molecular Imaging Research Center (MIRCen), Fontenay-aux-Roses, France; [2]Bioanalysis and Pharmacology of Bioactive Lipids Research Group, Louvain Drug Research Institute, UCLouvain, Université catholique de Louvain, Bruxelles, Belgium; [3]Department of Precision Medicine, School of Medicine & Surgery - University of Campania "Luigi Vanvitelli", Naples, Italy; [4]Department of Neurosciences, Reproductive and Odontostomatological Sciences, "Federico II" University of Naples, Naples, Italy

**Abstract** Cortical lesions represent a hallmark of multiple sclerosis and are proposed as a predictor of disease severity. microRNAs are suggested to be important players in the disease pathogenesis and the experimental autoimmune encephalomyelitis animal model. We implemented a mouse model recapitulating more closely the human pathology as it is characterized by both an autoimmune heterogeneity and the presence of cortical lesions, two parameters missing in experimental autoimmune encephalomyelitis. In our model, mice clustered in two groups displaying high or low clinical scores. Upon cortical cytokine injection, lesions appeared with a specific topography while cortical miRNA profiles were altered. These two features differed according to disease severity. We evidenced changes in miRNA regulators and targets suggesting that miRNA alteration had functional repercussions that could explain the differences in cortical lesions. This model represents a crucial tool for the study of both miRNA involvement and cortical lesion formation in disease pathogenesis.

**\*For correspondence:**
nicolaorefice819@gmail.com

[†]These authors contributed equally to this work
[‡]These authors also contributed equally to this work

**Present address:** [§]University of Wisconsin-Madison, Waisman Center, Madison, United States

**Competing interests:** The authors declare that no competing interests exist.

## Introduction

Although multiple sclerosis (MS) has been the subject of many pre-clinical and clinical studies in the last 50 years, its pathogenesis is still not completely understood. It is proposed to be the consequence of an interplay between genetic susceptibility and environmental factors (e.g. low vitamin D levels, smoking, Epstein-Barr virus infection). MS is a complex and heterogeneous disease presenting various degrees of inflammation, gliosis, and neurodegeneration leading to differences in clinical manifestations and severity between patients (*Paz Soldan and Rodriguez, 2002*). For decades, the focus in MS was on white matter lesions, however, progress in imaging put forth the relevance of lesions in the gray matter, particularly in the cerebral cortex (*Filippi and Rocca, 2019*). It is now established that these cortical lesions (CLs) represent a hallmark of MS (*Lucchinetti et al., 2011*; *Filippi et al., 2010*). CLs were suggested to be predictors of long-term severity in MS (*Filippi and Rocca, 2019*; *Filippi et al., 2010*; *Treaba et al., 2019*). Actually, the presence of CLs in patients with a clinically isolated syndrome is suggested as a confirmation for MS diagnosis (*Filippi et al.,*

2010). Accordingly, the assessment of CLs has been added to the diagnostic criteria of MS (*Thompson et al., 2018*).

Despite their importance, the factors and the pathological mechanisms that determine the presence and the localization of CLs remain elusive. This is in part due to the challenges of obtaining an appropriate animal model to study these phenomena. However, it has been suggested that meningeal inflammation and immune cell infiltration, characterized by the presence of T-cell infiltrates, B-cells, and macrophages play a role in the development of CLs (*Treaba et al., 2019*; *Howell et al., 2011*). Besides inflammation, oxidative stress leading to DNA damage, and degeneration of oligodendrocytes and neurons, could also be implicated in CL formation (*Fischer et al., 2013*).

Recently, the direct involvement of microRNAs (miRNAs) in MS pathogenesis (*Junker et al., 2011*), as well as in the pathogenesis of the experimental autoimmune encephalomyelitis (EAE) animal model has been put forth (*Thamilarasan et al., 2012*). miRNAs are small non-coding RNA that mediate the repression of messenger RNA translation and thereby refine protein expression levels. It is estimated that up to 60% of protein-coding genes are regulated by miRNAs (*Friedman et al., 2009*). Several studies support miRNAs as having a central role in inflammation and adaptive immunity, emphasizing the need to understand their variation and potential implication in MS, as this could improve our understanding of MS pathogenesis. Dysregulation of the miRNA profile was reported in the peripheral blood (*Keller et al., 2014*), in active and inactive lesions, and in normal-appearing white matter in MS patients (*Junker et al., 2009*). However, miRNA analysis in a mouse model that features CLs is still needed to improve our understanding of their potential role. Indeed, this would provide a tool to study miRNA expression at the early stage of CL formation which is not feasible in human patients or in the other models of MS.

Therefore, the aim of this study was twofold. First, we wanted to set up and characterize an EAE mouse model recapitulating more closely the key aspects of MS, specifically the heterogeneity of the immune response and the presence of CLs. Secondly, we analyzed the miRNome in this modified model of MS to identify interesting candidate miRNAs that could potentially be involved in the phenotype observed.

## Results

### EAE mouse model with both a heterogeneous immune response and cortical lesions

One of the classical mouse models of MS is the EAE model that is typically induced by subcutaneous administration of myelin-derived peptides (such as $MOG_{35-55}$ when working with C57BL/6JRj mice) in complete Freund's adjuvant (CFA) followed by intraperitoneal administration of pertussis toxin (PTX). PTX administration is commonly used as it leads to a more homogeneous immune response. However, MS is a heterogeneous disease. Thus, in order to obtain in mice a more heterogeneous autoimmune response and to assess the impact of PTX injection in the same setting, we induced EAE in mice using $MOG_{35-55}$ in CFA with or without PTX administration. As might be expected, mice that received the PTX declared the disease earlier than mice that did not (*Figure 1A*). Using a standard five-point EAE grading scale, we could separate the EAE mice that did not receive PTX into two distinct cohorts (*Figure 1A*): mice with a high clinical score (HIS; 53% of the immunized mice) and mice with a low clinical score (LIS; 47%). Importantly, there was no difference in disease onset between the HIS and LIS groups. EAE in mice typically presents with ascending paralysis consistent with spinal cord lesions. Accordingly, we could observe decreased fluoromyelin staining in the spinal cord of EAE mice (*Figure 1B*). We also found less lesions in the LIS group compared to the HIS group, and that the latter could not be distinguished from the PTX group. Indeed, the decrease in fluoromyelin staining was more marked in the HIS group compared to the LIS group, consistent with the higher clinical score (*Figure 1B*). To assess reproducibility and strengthen our findings, the same immunization protocol without PTX was conducted on a second independent set of mice (different suppliers and laboratories). Here again, we observed two distinct phenotypes, that is HIS and LIS, with a similar distribution (*Figure 1—figure supplement 1A*). This was accompanied by decreased Luxol Fast Blue-Cresyl Violet (LFB-CV) staining in the spinal cord of both HIS and LIS mice of this second cohort. This decrease was more marked in the HIS group compared to the LIS group (*Figure 1— figure supplement 1B*).

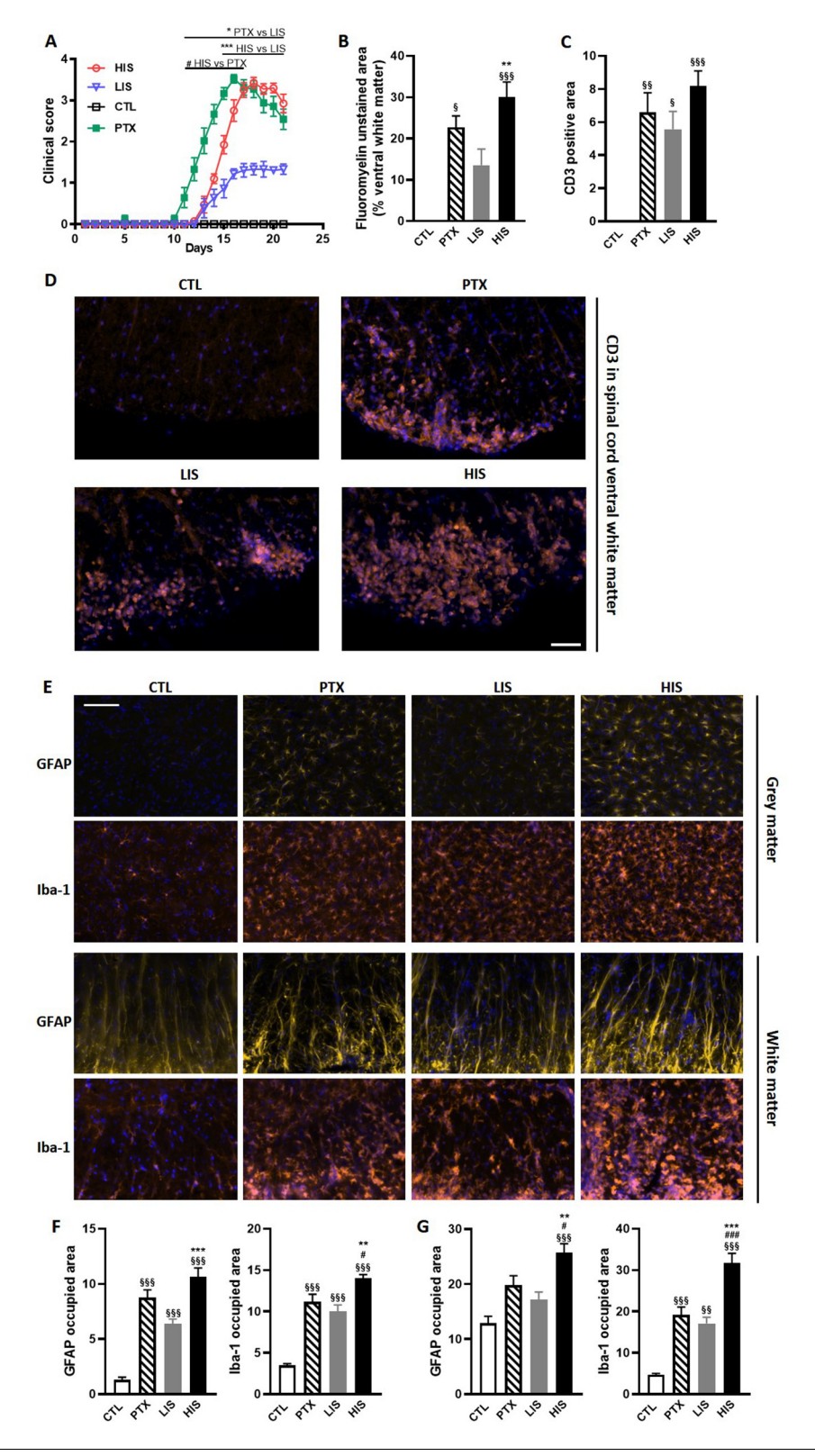

**Figure 1.** Clinical and immunohistological characterization of the EAE model with high and low clinical scores. To induce EAE, mice received MOG$_{35-55}$ and complete Freund's adjuvant (CFA) with or without pertussis toxin (PTX). (**A**) Clinical score of EAE mice receiving PTX (PTX group) or no PTX and with a high clinical score (HIS) or low clinical score (LIS). (**B**) Quantification of fluoromyelin signal in the spinal cord ventral white matter. (**C**) Quantification of the CD3 positive area in the ventral white matter of the spinal cord. (**D**) Representative photomicrographs of CD3 positive cells

*Figure 1 continued on next page*

*Figure 1 continued*

(lymphocytes) infiltrating the spinal cord ventral white matter. The scale bar represents 50 μm. The quantification of the entire cohort is shown in (C). (E) Representative photomicrographs of GFAP positive cells (astrocytes) and Iba-1 positive cells (microglia, monocytes, macrophages) in the spinal cord grey and white matters of mice. The scale bar represents 50 μm. (F-G) Quantification of GFAP positive area and Iba-1 positive area in (F) the grey matter and in (G) the white matter of the spinal cord. The scale bar represents the indicated length. Data are mean ± sem. N = 7–12/group. For A, two-way ANOVA with Sidak's post-hoc test. For B to F, one-way ANOVA with Sidak's post-hoc test. §$p \leq 0.05$ §§$p \leq 0.01$ §§§$p \leq 0.001$ vs CTL, #$p \leq 0.05$ ###$p \leq 0.001$ vs PTX, *$p \leq 0.05$ **$p \leq 0.01$ ***$p \leq 0.001$ vs LIS.

The online version of this article includes the following figure supplement(s) for figure 1:

**Figure supplement 1.** Spinal cord lesions are more pronounced in mice with high clinical score (HIS) compared to mice with low clinical score (LIS).

To further explore the differences between LIS and HIS, we first assessed CD3 immunoreactivity in the spinal cord as it reflects lymphocyte infiltration. CD3 immunostaining in all the mice that received the immunizing peptide was similar (*Figure 1C–D*). We then measured the area of the spinal cord occupied by Iba-1 (a marker of microglia, macrophages, and monocytes) and GFAP (a marker of astrocytes) positive cells. We found, both in the white and grey matters, larger Iba-1-occupied and GFAP-positive areas in the HIS group compared to the LIS group (*Figure 1E–G*). Next, we measured the mRNA expression of cytokines and chemokines as well as of the Treg transcription factor FoxP3 (*Foxp3*) and the Th17 transcription factor RORγ (*Rorc*) in the spinal cord. Immunization resulted in an increased mRNA expression of cytokines, chemokines, and *Foxp3* and a decreased expression of *Rorc* (*Figure 2A*). There were no differences between the HIS and LIS groups, except for *Foxp3* which was more expressed in the LIS group compared to the HIS group (*Figure 2A*). As the difference in disease development could also be explained by differences in blood cytokine levels, we measured at 7, 14 and 21 days post-immunization the levels of key inflammatory cytokines and chemokines (e.g. IL-17A, IFNγ, G-CSF, ...) (*Figure 2—figure supplement 1*). Circulating cytokine levels were similar in the HIS and LIS groups, except for IL-12p40 at day 14 and MIP-1α at day 21. Finally, we thought to assess whether differences between the HIS and LIS groups could also be evidenced in the cortex as it is the localization of the CLs found in MS patients. Therefore, we measured the mRNA expression of cytokines and chemokines and found increased mRNA expression of *Il1b*, *Tnf*, and *Cxcl10* in the immunized groups compared to the control group (*Figure 2B*). Interestingly, the cytokine *Tnf* and the chemokine *Cxcl10* were more expressed in the HIS group compared to the LIS group (*Figure 2B*). This was also the case for mRNA of the lymphocyte marker *Cd3g* (*Figure 2B*). When looking at the expression of neurotrophic factors, we found decreased expression of *Bdnf* in the immunized groups compared to the control group with no difference between HIS and LIS (*Figure 2C*). *Gdnf* expression on the other hand was increased only in the groups that did not receive PTX with higher expression in the HIS group compared to the LIS group (*Figure 2C*).

As mentioned, in humans, CLs are a key feature of MS pathogenesis. To induce these CLs in mice, we stereotactically injected TNFα and IFNγ into the primary somatosensory cortex of EAE mice (both HIS and LIS groups) on day 21 post-immunization and sacrificed them 3 days after (time needed for the potential CLs to form *Gardner et al., 2013*; *Merkler et al., 2006*). Stereotactic injection of PBS to EAE mice was also performed as a control. Of note, we obtained with this cohort a similar distribution of LIS (46%) and HIS (54%) mice compared to the previous cohorts. When analyzing the cortex of these mice, we found decreased LFB-CV staining in the ipsilateral cortices of both HIS-Cytokine (HIS-C) and LIS-Cytokine (LIS-C) mice but not in the EAE mice injected with PBS (HIS-PBS and LIS-PBS) (*Figure 3A–C*).

## Cytokine injection leads to different topography and extent of cortical lesions depending on EAE severity

The decreased LFB-CV staining was more marked in the HIS-C group compared to the LIS-C group (*Figure 3B*). These results were reinforced by fluoromyelin green staining. Indeed, fluoromyelin signal intensity was lower in the HIS-C group compared to the LIS-C group in the ipsilateral cortex (*Figure 3D,E*). This was further confirmed by lower cortical myelin basic protein (MBP) signal in the HIS-C group compared to the LIS-C (*Figure 3F*). Moreover, fluoromyelin signal intensity was not changed in the ipsilateral cortices of PBS injected groups compared to the control mice (*Figure 3B*). Finally, we assessed the topographic distribution of CLs. Indeed, the intracortical injection of

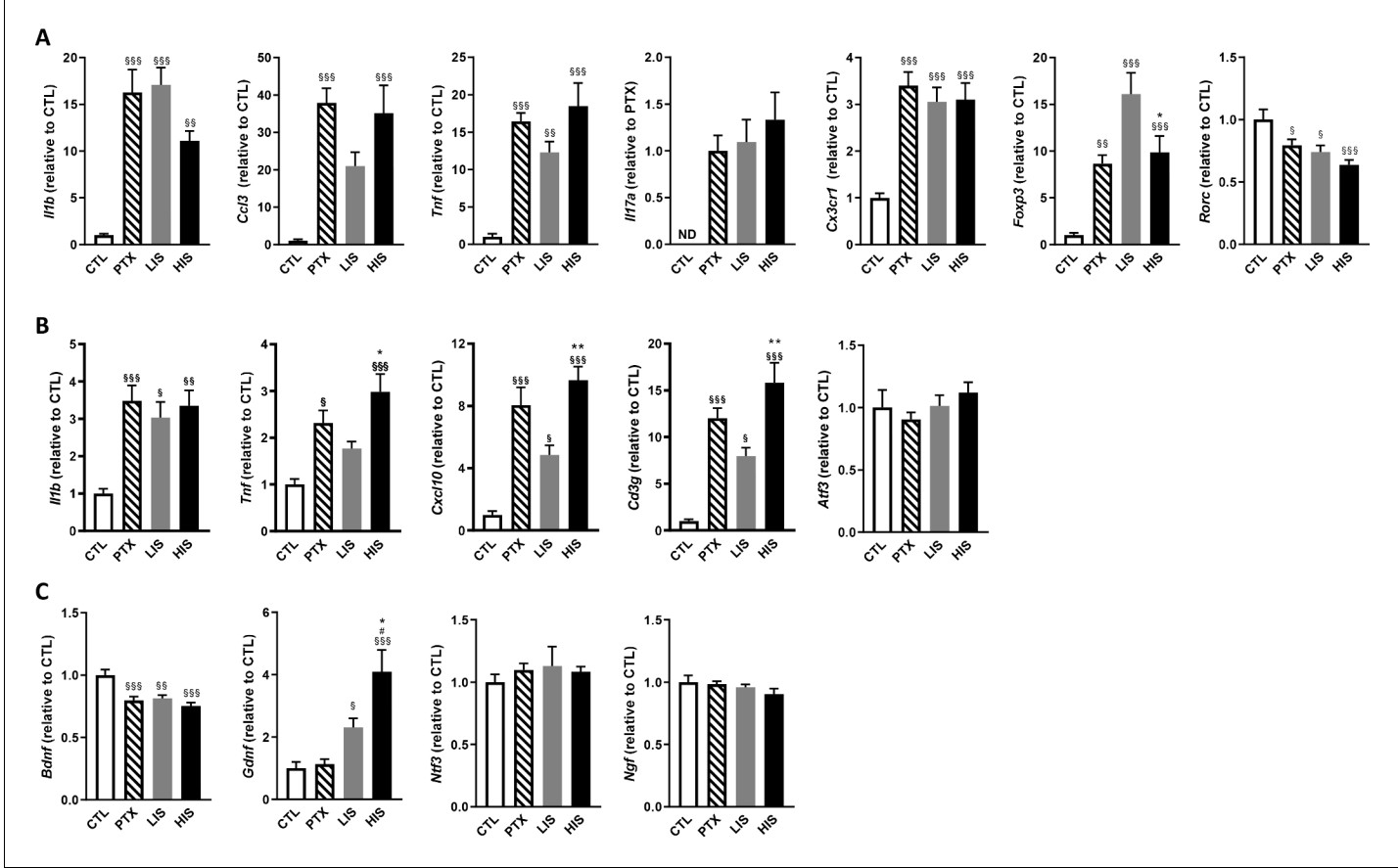

**Figure 2.** Characterization of inflammatory marker and growth factor expression in the EAE model with high and low clinical scores. To induce EAE, mice received MOG$_{35-55}$ and complete Freund's adjuvant (CFA) with pertussis toxin (PTX) or without PTX (HIS and LIS groups). (**A**) mRNA expression of *Il1b*, *Ccl3*, *Tnf*, *Il17a*, *Cx3cr1*, *Foxp3*, and *Rorc* was assessed by RT-qPCR in the spinal cord. (**B–C**) mRNA expression of *Il1b*, *Tnf*, *Cxcl10*, *Cd3g*, *Atf3*, and (**C**) the neurotrophins *Bdnf*, *Gdnf*, *Ngf*, and *Ntf3* was assessed by RT-qPCR in the cortex. Data are mean ± sem. The expression of the CTL group was set at 1, except for *Il17a* that was not detected in the control group (thus, the PTX group was set at 1). N = 7–12/group. One-way ANOVA with Sidak's post-hoc test. §$p \leq 0.05$ §§$p \leq 0.01$ §§§$p \leq 0.001$ vs CTL, # $p \leq 0.05$ vs PTX, *$p \leq 0.05$, **$p \leq 0.01$ vs LIS.

The online version of this article includes the following figure supplement(s) for figure 2:

**Figure supplement 1.** Time-course evaluation of inflammatory plasma cytokines and chemokines.

cytokines targeted the somatosensory cortex and we wondered if CLs could be found away from the injection site and if differences in their distribution could be found between mice from the HIS-C and LIS-C groups. In both groups, we evidenced a large number of CLs in the somatosensory cortex and in the motor and visual cortices albeit with a lower frequency than in the somatosensory cortex (*Figure 4A–C*). However, we found no difference in CL distribution between the HIS-C and LIS-C groups (*Figure 4A–C*). MBP-negative area in intracortical and subpial CLs was higher in the HIS-C group compared to the LIS-C group (*Figure 4D–F*).

Because astrocyte morphology in MS varies in accordance with the stage and lesion type, we performed GFAP staining in the ipsilateral cortex of HIS and LIS mice. GFAP staining in both HIS-C and LIS-C mice was increased around the lesion site, showing prominent astrogliosis (*Figure 3G*). Conversely, Iba-1 staining showed no difference between HIS-C and LIS-C mice (*Figure 4G*). To study lymphocyte recruitment, we assessed the presence of CD3 positive cells within the cortex of these mice. We found no difference between the HIS-C and LIS-C mice (*Figure 4—figure supplement 1*). In MS patients, demyelination can be accompanied by neuronal damage (*Lassmann, 2018*), therefore, we investigated neuronal loss using toluidine blue staining (*Victório et al., 2010*) and NeuN immunostaining (*Sato et al., 2001*). We found cortical areas with a decrease in toluidine blue

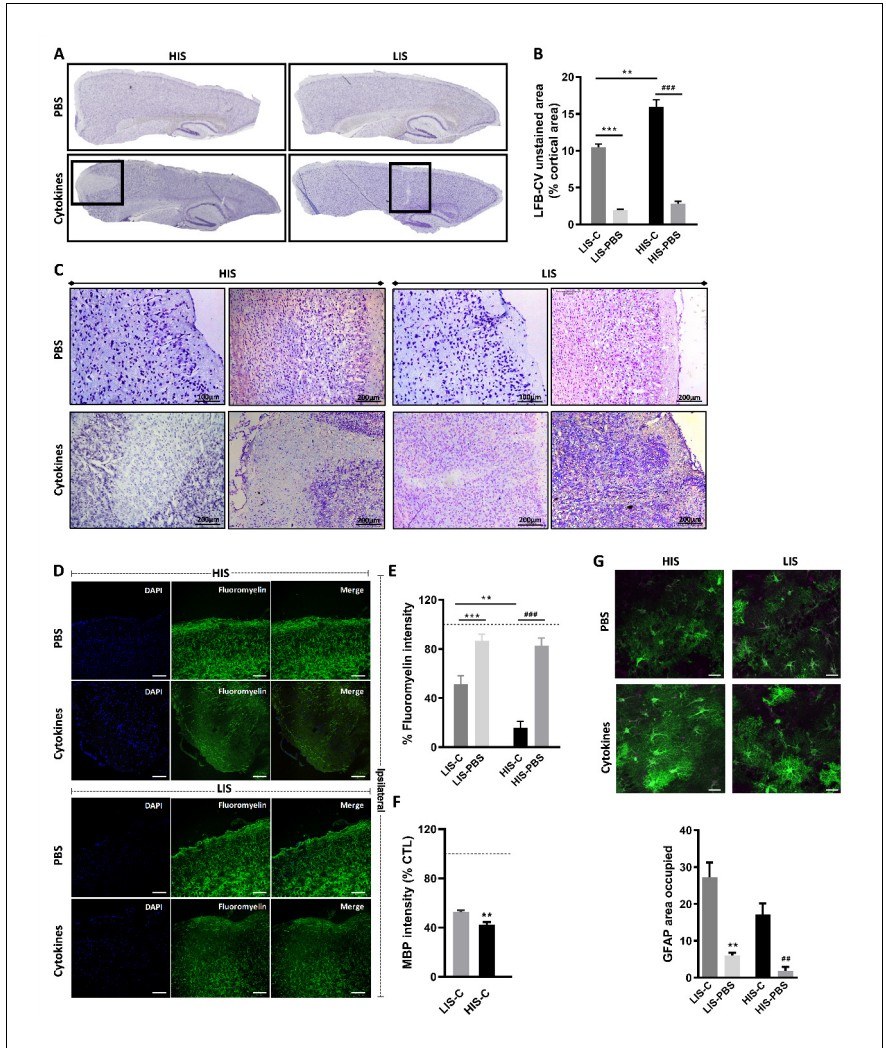

**Figure 3.** Cortical lesion extent depends on high and low clinical scores. (**A**) Representative photomicrographs of cortices stained with luxol fast blue and counter-stained with cresyl violet (LFB-CV) and showing the presence of CLs. (**B**) Quantification of LFB-CV unstained area in mice cortices with a high clinical score (HIS) or low clinical score (LIS) receiving either an injection of the cytokine (-C) or of PBS. (**C**) Representative photomicrographs depicting close-ups of the ipsilateral cortex of HIS and LIS mice displayed in panel A. (**D**) Representative photomicrographs and (**E**) quantification of fluoromyelin green intensity in the ipsilateral cortex of HIS-C and LIS-C mice. (**D**) The scale bar represents 100 µm. (**F**) Quantification of MBP intensity in the ipsilateral cortex of HIS-C and LIS-C mice. The dotted line represents the intensity of (**E**) fluoromyelin or (**F**) MBP immunostaining measured in CTL mice and set at 100%. (**G**) Representative photomicrographs and quantification of GFAP immunofluorescence in the ipsilateral cortex of HIS and LIS mice. The scale bar represents 20 µm. HIS-C: high clinical score mice injected with the cytokine mixture; HIS-PBS: high clinical score mice injected with PBS; LIS-C: low clinical score mice injected with the cytokine mixture; LIS-PBS: low clinical score mice injected with PBS. Data are mean ± sem. For B, one-way ANOVA with Dunnett's post-hoc test. For E and G, one-way ANOVA with Sidak's post-hoc test, **p≤0.01, ***p≤0.001 vs LIS-C and ## p≤0.01, ### p≤0.001 vs HIS-C. For F, two-tailed t-test **p≤0.01.

staining, but no difference between the HIS and LIS groups (*Figure 4—figure supplement 2*). Similar data were obtained for NeuN immunostaining (*Figure 4H*).

## Cortical lesion formation is not affected by the timing of cytokine injection

Next, we asked whether injecting the cytokines at a later time point in the HIS cohort would result in a different extent of CLs. Therefore, we immunized mice with MOG and CFA (but no PTX) and after

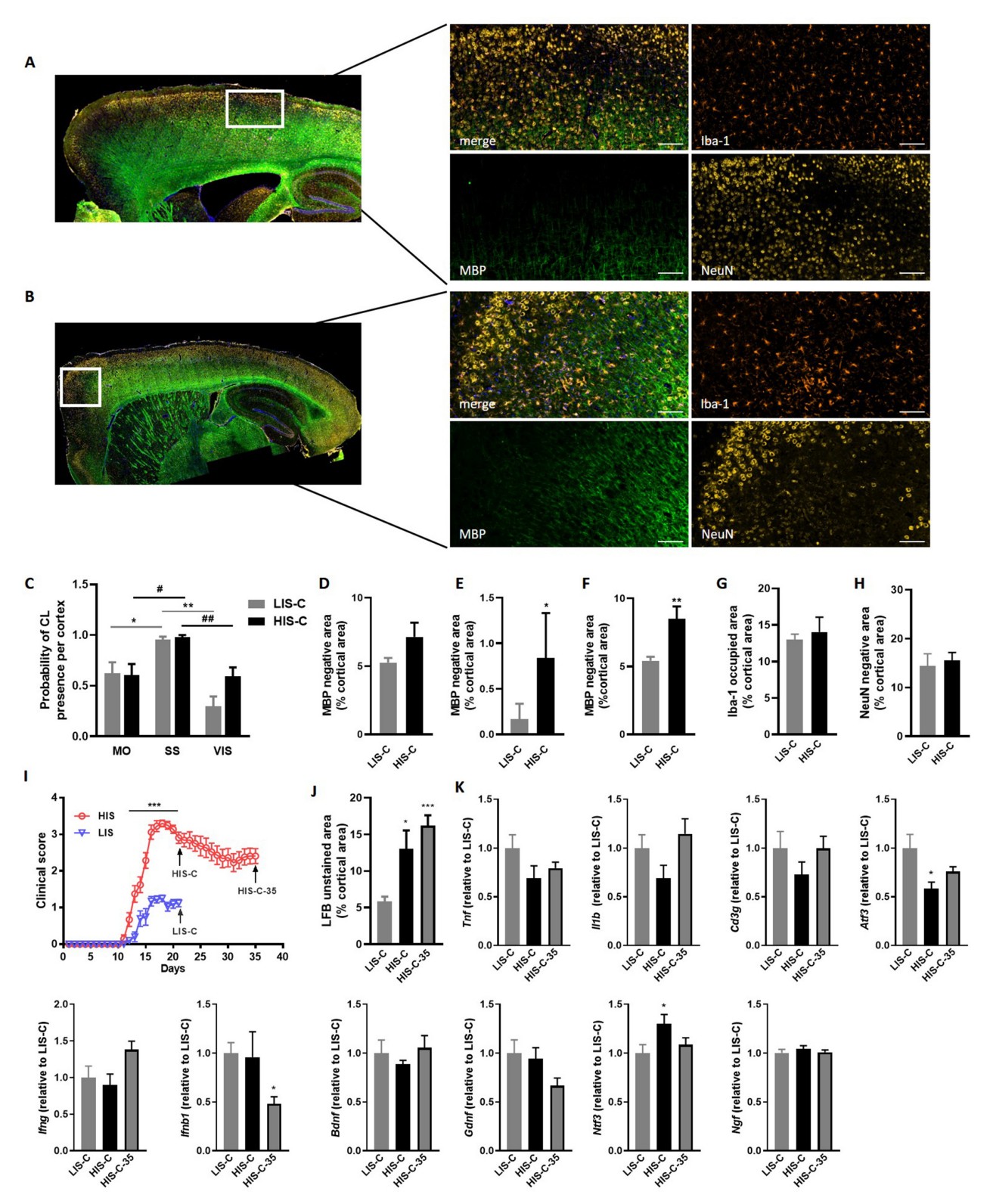

**Figure 4.** Cortical lesion distribution and extent differ between mice with a high clinical score (HIS-C) and with a low clinical score (LIS-C) after cytokine injection. Representative photomicrographs and close-ups of the ipsilateral cortex of (A) LIS-C and (B) HIS-C mice immunostained with Iba-1, MBP, and NeuN. The scale bar represents 100 µm. (C) Quantification of the probability of CL presence in the motor cortex (MO), the somatosensory cortex (SS), and the visual cortex (VIS) in the ipsilateral cortex of LIS-C and HIS-C mice (0:never present, 1:always present). Quantification of (D) intracortical, (E)

*Figure 4 continued on next page*

*Figure 4 continued*

subpial, and (**F**) total CL size in the ipsilateral cortex of LIS-C and HIS-C mice. (**G**) Quantification of Iba-1 occupied area in the cortex of HIS-C and LIS-C mice. (**H**) Quantification of NeuN negative area in the ipsilateral cortex of HIS-C and LIS-C mice. (**I**) Clinical score for another cohort of EAE mice with HIS and LIS groups that received cytokine injection either at day 21 post-immunization (HIS-C and LIS-C) or at day 35 post-immunization (HIS-C-35). All mice were euthanized three days after cytokine injection. (**J**) Quantification of LFB unstained area in cortices for the mice in panel I. (**K**) mRNA expression of *Tnf*, *Il1b*, *Cd3g*, *Atf3*, *Ifng*, *Ifnb1*, *Bdnf*, *Gdnf*, *Ntf3* and *Ngf* was measured by RT-qPCR in mice with low clinical score and injected with cytokines on day 21 (LIS-C), mice with high clinical score and injected with cytokines on day 21 (HIS-C), and in mice with high clinical score and injected with cytokines on day 35 (HIS-C-35). The expression of the LIS-C group was set at 1. Data are mean ± sem. N = 8–10/group. For C two-way ANOVA with Sidak's post-hoc test, between cortical regions for LIS-C mice *p≤0.05, **p≤0.01; and for HIS-C mice #p≤0.05, ##p≤0.01. For D-H, two-tailed t-test *p≤0.05, **p≤0.01. For me, two-way ANOVA with Sidak's post-hoc test, ***p≤0.001. For J-K, one-way ANOVA with Sidak's post-hoc test *p≤0.05, ***p≤0.001 vs LIS-C.

The online version of this article includes the following figure supplement(s) for figure 4:

**Figure supplement 1.** Lymphocyte infiltration and microglia activation are similar in mice with high clinical score (HIS-C) and with low clinical score (LIS-C) after cytokine injection.

**Figure supplement 2.** Neuronal loss evidenced by toluidine blue is similar in mice with high clinical score (HIS-C) and with low clinical score (LIS-C) after cytokine injection.

21 days, we injected the cytokines in the cortex of the LIS mice and half the cohort of HIS mice (similarly to the previous study). The remaining mice received the cytokine injection at day 35 when their clinical score reached a plateau supporting the fact that the mice had reached a chronic phase of the disease (*Figure 4I*). Interestingly, the extent of LFB staining loss was not affected by the timing of cytokine injection (*Figure 4J*). Furthermore, mRNA expression of the cytokines *Il1b*, *Tnf*, and *Ifng* and expression of the lymphocyte marker *Cd3g* were not different between the three groups (LIS-C, HIS-C, HIS-C-35) (*Figure 4K*). However, the expression of *Atf3* (activating transcription factor 3), a marker of cellular stress, was lower while the expression of the neurotrophin *Ntf3* was higher in the HIS-C group compared to the LIS-C group (*Figure 4K*). Concerning the HIS-C-35 group, we only found a decrease in *Ifnb* mRNA expression compared to the other two groups (*Figure 4K*). Of note, several markers that were increased in the cortex when comparing HIS mice and LIS mice (namely *Cxcl10*, *Tnf*, *Cd3g*, and *Gdnf*) were not altered in HIS-C versus LIS-C mice (*Figures 2B* and *4K*). Therefore, we decided to compare the expression levels of these various markers before and after cytokine injection (*Figure 5*). Our data show that cytokine injection increased the mRNA expression of *Il1b*, *Tnf*, *Cxcl10*, *Atf3*, *Ngf*, and *Ntf3* in both LIS and HIS groups (*Figure 5*). Conversely, *Bdnf* was decreased in LIS-C compared to LIS mice while *Gdnf* was decreased in HIS-C compared to HIS mice (*Figure 5*). Finally, *Cd3g* mRNA expression was increased in the LIS-C group compared to the LIS group but not in the HIS-C group (*Figure 5*), suggesting that cytokine injection increased lymphocyte infiltration in the LIS group.

Our data so far further support the fact that stereotactic injection of cytokines in the cortex of EAE mice induces the formation of CLs. Moreover, in our model, CL features were dependent on the clinical score but not on cytokine injection timing. Additionally, cytokine injection seemed to dampen some of the differences found between the LIS and HIS groups.

## EAE mice injected with cytokines show a dysregulated miRNA profile at the injection site

Next, we asked whether miRNA expression in the cortex of HIS and LIS mice could be differentially affected by the injection of cytokines. Indeed, miRNAs are important players in the control of inflammation and immune responses and their characterization has not been carried out in a model displaying CLs. Thus, we performed a high-throughput miRNA expression profile using TaqMan microfluidic cards on the micro-dissected cortical region surrounding the injection site and the corresponding contralateral structure of HIS-C and LIS-C mice. We found 202 miRNAs expressed in the ipsilateral and 197 miRNAs in the contralateral side of HIS-C mice, 197 miRNAs in the ipsilateral, and 179 miRNAs in the contralateral of LIS-C mice, and 167 miRNAs expressed in control mice (i.e. PBS-injected non EAE mice, CTL). A subset of 155 miRNAs was expressed in all samples (*Supplementary file 1*).

Next, we compared the miRNA expression profiles in these different groups with the control mice (*Supplementary file 2*). For miRNA analysis, an increase was considered when miRNA

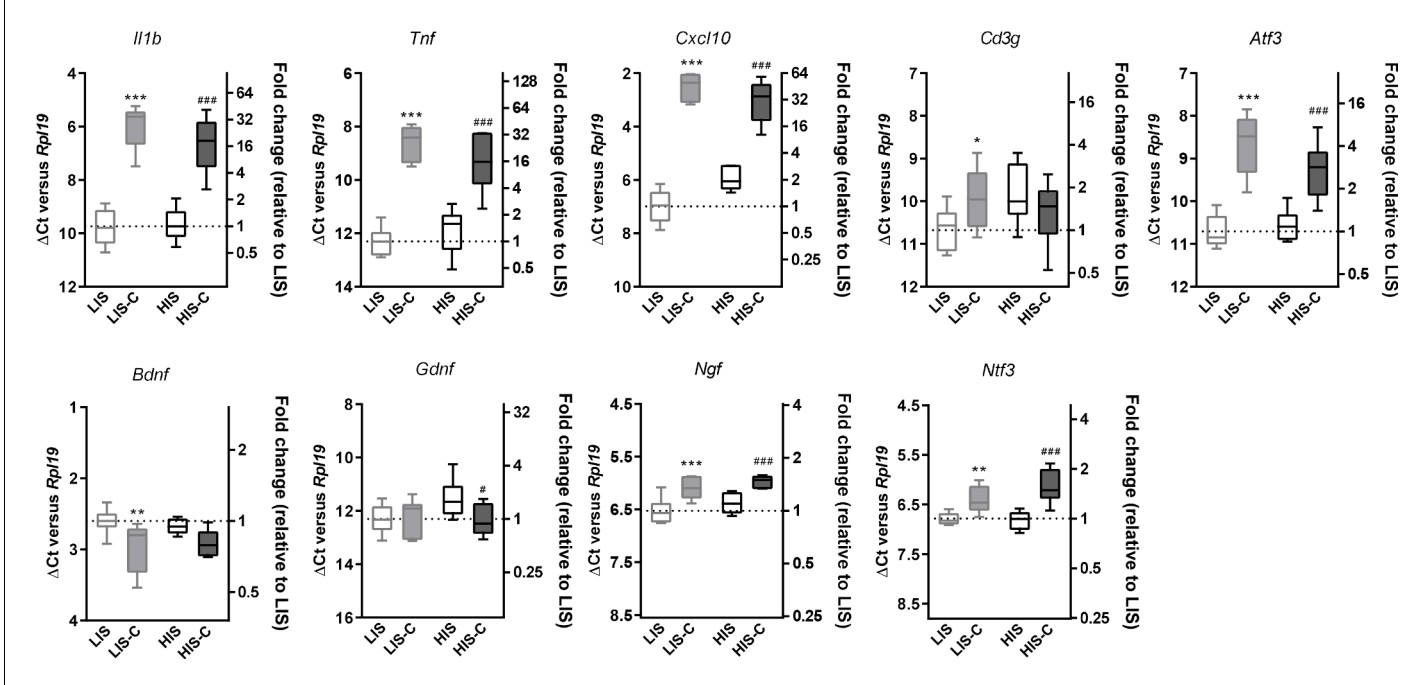

**Figure 5.** Alterations in mRNA expression in the ipsilateral cortex before and after cytokine injection. mRNA expression of *Il1b*, *Tnf*, *Cxcl10*, *Atf3*, *Cd3g*, *Bdnf*, *Gdnf*, *Ngf* and *Ntf3* measured by RT-qPCR in the ipsilateral cortex of mice with a low clinical score before (LIS) and after (LIS-C) cytokine injection and in mice with a high clinical score before (HIS) and after (HIS-C) cytokine injection. Data are plotted as ΔCt between the gene considered and the reference gene used (*Rpl19*) (left vertical axis) and as the fold-increase of the expression between LIS group (set at 1) and the three remaining groups (right vertical axis). The smaller the ΔCt between the gene considered and *Rpl19*, the more the gene is expressed, and conversely, the bigger the ΔCt the less the gene is expressed. Data are box (median) and whiskers (min to max). One-way ANOVA, *p≤0.05, **p≤0.01, ***p≤0.001 vs LIS, #p≤0.05, ###p≤0.001 vs HIS.

expression was ≥2 fold and a decrease was considered when miRNA expression was ≤0.5. Analysis of the ipsilateral injected structure of HIS-C versus CTL showed 161 miRNAs were expressed in both groups and among them, 139 were upregulated, and none were downregulated. For the corresponding contralateral cortical structure of HIS-C, 161 miRNAs were expressed in both HIS-C and CTL of which 138 were upregulated and two were downregulated. Analysis of the ipsilateral injected structure of LIS-C versus CTL showed 160 miRNAs expressed in both groups and among them, 127 were upregulated, and four were downregulated. For the corresponding contralateral cortical structure of LIS-C, 157 miRNAs were expressed in both LIS-C and CTL of which 115 were upregulated and two were downregulated. These data suggest that, in the cortical region surrounding the injection site, EAE induces miRNA expression compared to the control group as more miRNAs were expressed in the HIS-C and LIS-C groups compared to CTL, and many of the miRNAs expressed in all groups were up-regulated compared to the control group (*Supplementary file 2*).

As we were interested in the difference between HIS-C and LIS-C groups, we compared the miRNA profiles between the ipsilateral cortical region surrounding the injection site of HIS-C and LIS-C mice (*Supplementary file 3*). We found 192 miRNAs expressed, 20 of which were more expressed in HIS-C, and eight were less expressed. miRNAs altered in HIS-C compared to LIS-C are shown in *Figure 6A*. Some miRNAs were expressed only in HIS-C or LIS-C mice and are reported in *Figure 6B*. We also compared the miRNA profiles between the contralateral corresponding cortical structure of HIS-C and LIS-C mice (*Supplementary file 3*). In this case, we found 177 miRNAs expressed, 59 of them were more expressed in HIS-C compared to LIS-C while only one miRNA was less expressed. In silico analysis on the miRNAs that were altered in HIS-C-ipsi vs LIS-C-ipsi or were found only in either HIS-C-ipsi or LIS-C-ipsi predicted the involvement of these microRNAs in several pathways known to be involved in MS (*Figure 6C*) including neurotrophin signaling, FoxO signaling and T cell signaling. When analyzing the pathways that were predicted to be involved by all three of

the databases we used, we found 13 miRNAs that were mostly found in these pathways and controlling the most genes (*Figure 6D*). These miRNAs control an array of genes and shown is a panel of the targeted genes found in at least two of the three databases used (*Figure 6D*). Some of these genes are generally involved in signaling pathways or in cancer but some others, such as *Mbp* could point to a role for these miRNAs in the pathogenesis of EAE. Indeed, four of the altered miRNAs (*Mir152-3p*, *Mir7b-5p*, *Mir148a-3p*, and *Mir7a-5p*) had *Mbp* as target gene (*Figure 6D*) and could, therefore, play a role in myelination.

## *Mir155* and *Mir223* and their target FOXO3 are differently altered depending on EAE severity

Another interesting gene in these pathways is the gene for the transcription factor FOXO3, which could be targeted by three of the differentially altered miRNAs: *Mir155-5p*, *Mir223-3p*, and *Mir29b-3p*. Indeed, while this transcription factor is not yet extensively studied in the context of MS, FOXO3 controls Th1 cell differentiation, inhibits oligodendrocyte progenitor cell differentiation, and was shown to exert an important role in neuroinflammation (*Stienne et al., 2016*; *Srivastava et al., 2018*; *Stefanetti et al., 2018*). Moreover, *Foxo3*-deficient mice exhibit reduced susceptibility to EAE (*Stienne et al., 2016*). Here, we observed that *Mir29b-3p* was expressed in the ipsilateral cortical structure surrounding the injection site of LIS-C-ipsi but not in HIS-C-ipsi. *Mir155-5p* and *Mir223-3p* were upregulated in the injection site of HIS-C and LIS-C groups compared to CTL. The upregulation was stronger in LIS-C than in HIS-C for *Mir155-5p* but the opposite was true for *Mir223-3p* (*Supplementary file 1*). Therefore, when comparing HIS-C-ipsi versus LIS-C-ipsi, we found *Mir155-5p* to be less expressed in HIS-C, while *Mir223-3p* was more expressed. Interestingly, the profile of *Mir155-5p* was reversed in the corresponding contralateral structure as it was more expressed in HIS-C than in LIS-C (*Supplementary file 3*). The expression of these miRNAs was also assessed using real-time PCR (*Figure 7A*). *Mir155-5p* and *Mir223-3p* are known to be important in the context of neuroinflammation and MS and to be up-regulated in active white matter lesions from MS patients, however, their profile has never been studied at the site of the initial inflammatory process (*Junker et al., 2009*; *Chen et al., 2018a*; *Fenoglio et al., 2012*; *Murugaiyan et al., 2011*; *Zhang et al., 2014*; *Escobar et al., 2014*). Therefore, we studied whether these expression changes could translate into target protein changes. Here we found a lower FOXO3 immunofluorescence signal in the ipsilateral cortex of LIS-C mice compared to their contralateral cortex and compared to HIS-C mice (*Figure 7B*). However, no other differences in FOXO3 quantification were found. Therefore, the higher expression of miR-155 found in the ipsilateral cortex of LIS-C mice could be consistent with the lower FOXO3 expression. Moreover, we measured mRNA expression of *Cxcl10*, a direct target of miR-223–3 p, and found its expression to be lower in the HIS-C group (*Figure 7C*) consistent with the increased *Mir223-3p* expression. Finally, we found a decreased *Foxp3* mRNA expression consistent with the decrease in *Mir155-5p* as this miRNA is a direct target of FoxP3 (*Zheng et al., 2007*; *Figure 7C*). Contrary to the Treg transcription factor FoxP3, expression of the Th17 transcription factor *Rorc* was increased in HIS-C compared to LIS-C (*Figure 7C*). This suggests a difference in T cell polarization between HIS-C mice and LIS-C mice.

## C1q and its target *Mirlet7c* are differently altered depending on EAE severity

Interestingly, *Mirlet7c-5p*, another miRNA that was put forth in our analysis and that is involved in inflammatory processes (*Yu et al., 2016*), is less expressed in the ipsilateral cortex of HIS-C mice compared to LIS-C (*Figure 6A* and *Figure 7A*). *Mirlet7c-5p* was shown to be regulated by the complement's factor C1q (*Benoit and Tenner, 2011*), which was found to be activated in MS CLs (*Watkins et al., 2016*). Thus, we assessed C1q immunofluorescence levels in CLs in the ipsilateral and contralateral cortices of HIS-C and LIS-C mice. Consistent with the expression of *Mirlet7c*, the percentage of C1q-positive cells was higher in ipsilateral CLs of HIS-C compared to LIS-C mice and compared to their own contralateral cortex (*Figure 7D*). Interestingly, *Ntf3* mRNA expression, which was described to be controlled by *Mirlet7c* (*Benoit and Tenner, 2011*), is higher in the HIS-C mice compared to the LIS-C mice. These data further reinforce the strength of our model as mice displayed CLs and recapitulated another hallmark of MS, the activation of the complement pathway (*Watkins et al., 2016*).

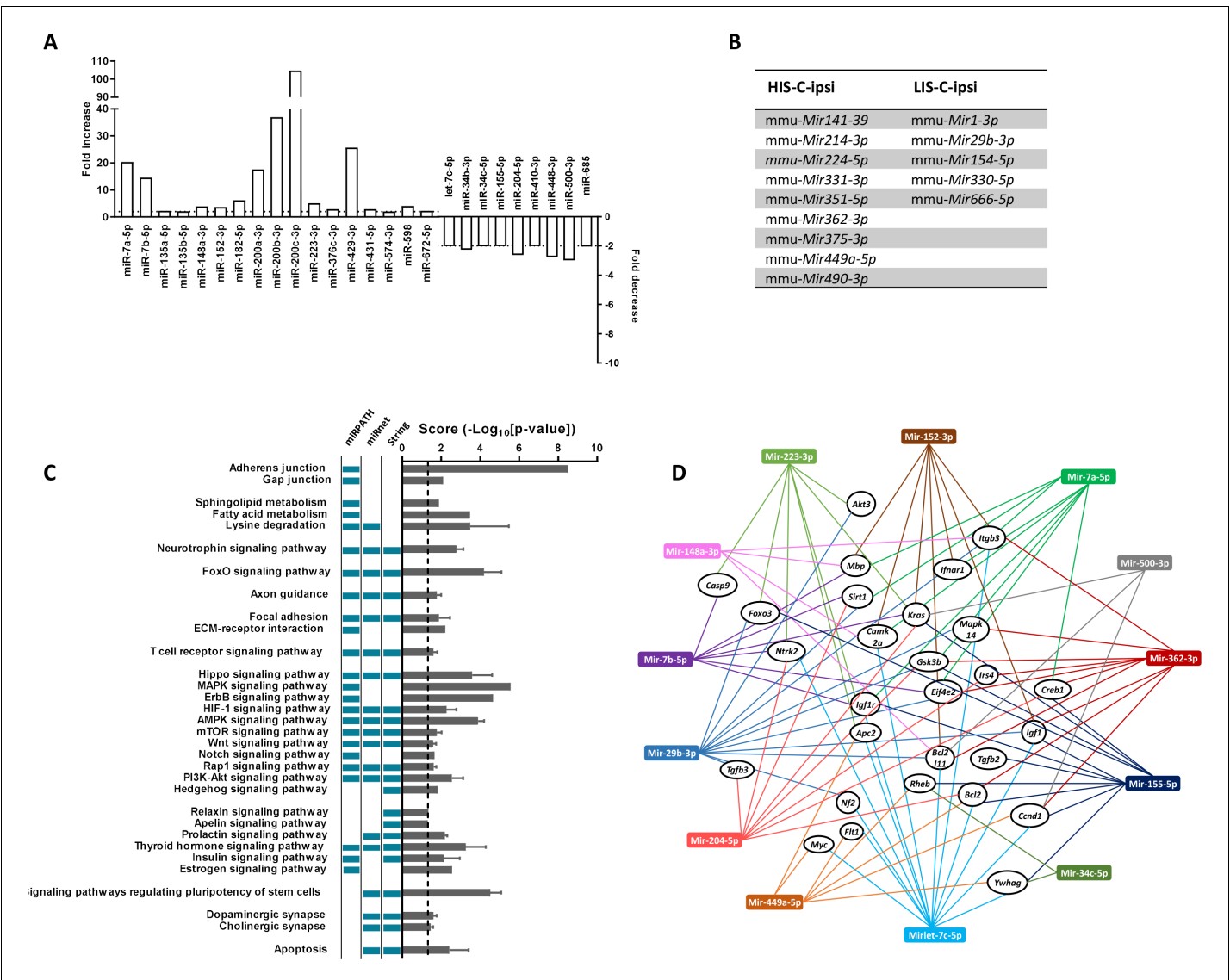

**Figure 6.** miRNA profiling and corresponding prediction of signaling pathway alterations in LIS-C and HIS-C mice. miRNA profile was assessed using TaqMan Microfluidic Array Cards Type A in control mice and HIS or LIS mice receiving a stereotactic injection of cytokines. (**A**) Variations of miRNAs at the injection site of EAE mice with high clinical score receiving cytokine injection (HIS-C-ipsi) compared to the injection site of EAE mice with low clinical score receiving cytokine injection (LIS-C-ipsi). Two-fold change for upregulation (plotted on the left y-axis) and downregulation (plotted on the right y-axis) are indicated by the dotted lines. Complete miRNA variations between the HIS-C and LIS-C groups are found in *Supplementary file 3*. (**B**) miRNAs expressed in HIS-C-ipsi but not in LIS-C-ipsi and inversely. (**C**) Predicted functional analysis of the top enriched canonical pathways associated with the target genes related to the miRNAs dysregulated in HIS-C-ipsi vs LIS-C-ipsi and found either in HIS-C-ipsi or LIS-C-ipsi. Indicated pathways were found by performing the Kyoto Encyclopedia of Genes and Genomes analysis by miRNet, mirPath, or Cytoscape String app databases. The pathways retrieved by specific databases are indicated by the colored boxes to the left of the graph (unchecked boxes represent pathways not found in the labeled database). The results are expressed as mean ± sem of -log10 adjusted p-value between the three databases; the dotted line designates the threshold of 1.3 (representing p-value at 0.05). (**D**) Schematic network representation of interactions between the 13 miRNAs that were mostly found in the pathways represented in C (only the pathways found in all three databases) and controlling the most genes and a panel of the targeted genes found in at least two databases.

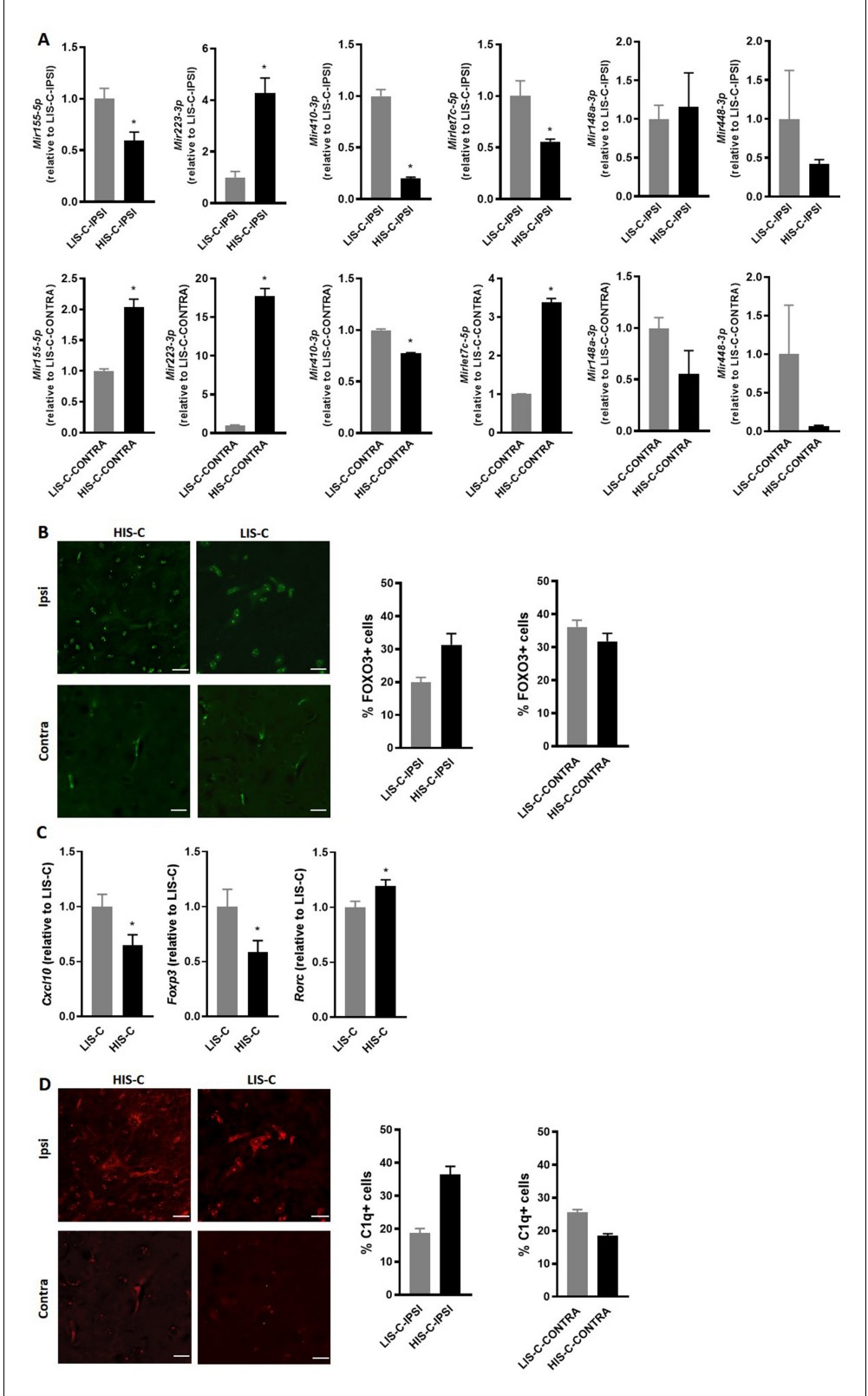

**Figure 7.** Alterations in miRNA levels are associated with variations of their regulator and targets. (**A**) *Mir155-5p*, *Mir223-3p*, *Mir410-3p*, *Mirlet7c-5p*, *Mir148a-3p* and *Mir448* expression was measured by RT-qPCR in the injection site and the corresponding contralateral structure of mice receiving cytokine injection with a high clinical score (HIS-C) and low clinical score (LIS-C). U6 was used as a reference. LIS-C ipsilateral (upper panels) or LIS-C contralateral (lower panels) levels were set at 1. N = 4–5/group. Data are presented as mean ± sem. *p≤0.05. (**B**) Representative confocal

*Figure 7 continued on next page*

*Figure 7 continued*

photomicrographs and quantification of FOXO3 (green) in HIS-C and LIS-C groups for both ipsilateral and contralateral sides of the cortex detected by immunofluorescence N = 2/group. (**C**) mRNA expression of Cxcl10, Foxp3, and Rorc was assessed by RT-qPCR in the ipsilateral cortex of HIS-C and LIS-C groups. (**D**) Representative confocal photomicrographs and quantification of C1q (red) in HIS-C and LIS-C groups for both ipsilateral and contralateral sides of the cortex detected by immunofluorescence N = 2/group. N = 9/group. Data are mean ± sem. The expression of the LIS-C group was set at 1. Two-tailed t-test, *p≤0.05.

## Discussion

Cortical demyelination has been evidenced post-mortem in patients suffering from MS (*Lassmann, 2012*). This process has also been suggested to have pathophysiological relevance because the white matter demyelination cannot explain by itself the broad range of clinical signs associated with MS (*Kutzelnigg et al., 2005*). Over the years, using new MRI sequences, cortical lesions were frequently found and are now identified as a hallmark of the pathology and are associated with inflammation (*Kutzelnigg et al., 2005*; *Gh Popescu and Lucchinetti, 2012*). However, the driving factors involved in these processes are still unknown. In our study, we wanted to set up in mouse a model able to recapitulate two key aspects of MS pathogenesis: the immune response heterogeneity and the presence of CLs. Indeed, such a model would allow us to study and mechanistically address these events at early stages of the disease. The first aspect was achieved by omitting the usual PTX administration in the classical EAE mouse model. Indeed, MS is a disease characterized by a wide heterogeneity in its clinical symptoms and its course. Our model can mimic this MS hallmark as we found in several independent studies (mice, laboratories, researchers) that the clinical score of the mice was no longer leveled out but could be stratified in two groups of similar size. This heterogeneity might seem surprising for inbred mice. However, inter-individual phenotypic variability has been reported for inbred mice in several models. In a two-bottle choice alcohol drinking paradigm, C57Bl/6 mice can be differentiated in low and high alcohol drinkers (*Juarez et al., 2017*; *Wolstenholme et al., 2011*). In a model of social defeat, C57Bl/6 mice can be clustered into susceptible and unsusceptible groups based on their behavior following chronic social defeat (*Krishnan et al., 2007*). In models of diet-induced obesity, C57BL/6 mice can be clustered in low and high weight-gainers as well as based on their insulin resistance phenotype (*Koza et al., 2006*; *Chen et al., 2018b*). However, in most experimental paradigms, the reasons for these differences remain to be fully explored, even if epigenetic mechanisms have been suggested to play a role in the observed variations. Actually, epigenetics could also explain differences in disease susceptibility between monozygotic twins (*Fraga et al., 2005*). Although the prevalence of MS is higher in monozygotic twins compared to the general population, both twins do not necessarily develop MS. This observation highlights both the role of genetic susceptibility in MS pathogenesis and the importance of environmental factors and epigenetic modifications (*Vakhitov et al., 2020*). Indeed, epigenetic mechanisms, such as DNA methylation and histone modifications as well as miRNAs have been proposed to be involved in MS pathogenesis (*Vakhitov et al., 2020*).

A second key aspect of MS is the presence of CLs which is not reported in the classical EAE model. We were able to induce this key feature in our EAE mouse model, similar to what was reported in an EAE model in common marmoset and rats (*Gardner et al., 2013*; *Merkler et al., 2006*; *Stassart et al., 2016*; *Üçal et al., 2017*). We induced CLs by the stereotactic injection of TNF$\alpha$ and IFN$\gamma$ in the mouse cortex. This additional inflammatory insult, added to the MOG$_{35-55}$ immunization, led to the formation of CLs that was not found in PBS-injected animals. The decrease in fluoromyelin, LFB, and MBP cortical staining was dependent on the clinical score of the mice as it was different in HIS-C and LIS-C mice but did not differ between HIS mice injected at the peak of disease (d21) or during its chronic phase (d35). We found CLs to be intracortical and to a lower extent subpial hence sharing close similarities to CLs observed in patients suffering from MS (*Lassmann, 2018*). In a model using rats immunized with rMOG and cortically injected with cytokines, *Merkler et al., 2006* also observed CLs but they did not report any heterogeneity of the autoimmune response as evidenced by the levels of circulating anti-MOG IgG titers. In our study, we were able to repeat this finding in mice and we also showed that some CLs were quite distant from the injection site such as those we observed in the motor cortex. In an interesting experiment, *Rüther et al., 2017* combined both EAE and cuprizone models in mice and observed CLs

characterized by microglia activation and monocyte recruitment. However, in this dual model, there is no heterogeneity reported in the autoimmune response.

As previously stated, the processes underlying cortical demyelination in MS remain to be determined. In our model, we evidenced the presence of CLs and we started to characterize them in the HIS-C and LIS-C groups. It would be interesting to characterize these CLs further in terms of myelin and axonal losses (e.g. using electron microscopy) or regarding the specific type of immune cells recruited (T-cell subtypes, monocytes/macrophages, . . .). A follow-up study should address these interesting aspects.

Most of the markers we measured in the brain before cytokine injection were increased in all the immunized groups. An exception was *Bdnf* that was decreased by 20%. When comparing the HIS and LIS groups, we found a significantly higher expression of the chemokine *Cxcl10* and the lymphocyte marker *Cd3g* in the HIS group. The comparison of the expression level in the cortex of mice that did not receive the cytokines and those injected with cytokines points to an upregulation of the expression of several key markers in the context of MS (*Figure 5*). For instance, the injection of cytokines resulted in increased cortical expression of *Il1b*, *Tnf*, and *Cxcl10* in both LIS and HIS groups. miRNAs and their effects are attracting attention as they represent important repressors of RNA translation and are now also considered as biomarkers and as potential therapeutic targets (*Irizar et al., 2015*; *Perdaens et al., 2020*). The dysregulation of miRNA profile is studied in the context of MS pathology, although mostly in blood (*Thamilarasan et al., 2012*; *Irizar et al., 2015*; *Jagot and Davoust, 2016*). A limited number of studies have profiled miRNA expression in the central nervous system (CNS) of deceased MS patients revealing that miRNAs are altered in MS lesions (*Junker et al., 2009*; *Wu et al., 2015*; *Noorbakhsh et al., 2011*). In these conditions, it is not possible to link the dysregulation of miRNA expression and early events in the formation of CLs. Thus, to further support the interest of our model, we wondered if the different clinical scores and CLs would result in differential changes in miRNA expression. Our results clearly show this is the case as the miRNome is strongly affected by the injection of cytokines, and several miRNAs are differently altered depending on the HIS or LIS group. We also focused on some miRNAs of particular interest in the disease. Among those, miR-155 is a major regulator of inflammation. The altered *Mir155* profile observed in our modified EAE model, along with prominent astrogliosis, might reflect the presence of an active lesion state in the LIS-C group. Interestingly, Junker et al found increased *Mir155* expression in astrocytes from active lesions of MS patients (*Junker et al., 2009*). Besides, the administration of miR-155 antagomir reduced the disease severity when administered in the classic EAE mouse model (*Murugaiyan et al., 2011*).

Another miRNA found dysregulated in many inflammatory conditions is miR-223 (*Haneklaus et al., 2013*). Here, we found *Mir223* to be consistently more expressed in mice from the HIS-C group compared to mice from the LIS-C group for both the ipsi- and contralateral cortical areas. We also found a lower expression of *Cxcl10*, one of miR-223 direct targets (*He et al., 2019*), in the HIS group. This glia-enriched miRNA has been studied in an EAE model, and *Mir223* KO mice were found to develop less severe hallmarks of the disease (*Jovičić et al., 2013*; *Cantoni et al., 2017*). The authors also observed alterations of the immune system that could explain the partial protection against the EAE phenotype. Indeed, they found an increase in myeloid-derived suppressor cell number in the CNS along with a decrease in T-cell proliferation and associated neuroinflammation (*Cantoni et al., 2017*). This detrimental role exerted by miR-223 in the context of EAE could participate in the different cortical lesions observed between HIS-C and LIS-C mice.

We also found *Mirlet7c* levels to be decreased in HIS-C compared to LIS-C mice in the ipsilateral cortical side. This miRNA is known to be involved in neuroinflammatory processes as it both decreases microglia activation and exerts neuroprotective effects (*Ni et al., 2015*; *Lv et al., 2018*). As the decreased expression was restricted to the ipsilateral side (increased in the contralateral side), differential regulation of *Mirlet7c* could participate in the difference in CL pattern between HIS and LIS groups. Finally, the effects and possible involvement of miR-410 described as being enriched in neurons compared to the other CNS cells (*Jovičić et al., 2013*), in the observed phenotype remain less clear as it was studied in the CNS only with respect to its behavioral functions and in tandem with miR-379 (*Marty et al., 2016*). However, in the context of another autoimmune disorder where T-cells also represent a key effector, namely systemic lupus erythematosus, miR-410 levels were decreased in patients' T-cells compared to healthy controls (*Liu et al., 2016*). Moreover, the authors showed that the signal transducer and activator of transcription-3 (STAT-3) is a target of

miR-410. This finding is quite interesting in the context of EAE as inhibiting the STAT-3 pathway, which is central in T-cell differentiation, has proven to be beneficial (*Hou et al., 2017*).

In this study, we established a new model of EAE in mice that is closer to human pathology as it recapitulates the usual hallmarks of EAE with the addition of heterogeneity in the immune response (HIS and LIS groups) as well as the presence of CLs. We also showed that intracortical cytokine injections in mice with different immune responses and clinical scores results in different alteration of the miRNome. While the direct link between these differences in miRNA expression and CLs formation remains to be established, we put forth the heterogeneity in the response to immunization and the formation of CL as key features of our model.

# Materials and methods

**Key resources table**

| Reagent type (species) or resource | Designation | Source or reference | Identifiers | Additional information |
|---|---|---|---|---|
| Strain, strain background (female mice) | *Mus musculus*, female, C57BL/6JRj | Janvier Labs | SC-C57J-F | |
| Antibody | anti-mouse MBP (Chicken polyclonal) | Abcam | Ab123499 | (1:1000) |
| Antibody | anti-mouse FOXO3 (rabbit polyclonal) | Abcam | Ab177487 | (1:300) |
| Antibody | anti-mouse C1q (Mouse monoclonal) | Abcam | Ab71940 | (1:100) |
| Antibody | anti-GFAP (Rabbit polyclonal) | Dako | Z0334 | (1:1000) |
| Antibody | anti-mouse Iba-1 (rabbit polyclonal) | Wako | 019–19741 | (1:500) |
| Antibody | anti-mouse CD3 (rabbit polyclonal) | Dako | A0452 | (1:300) |
| Antibody | anti-NeuN (mouse monoclonal) | Millipore | MAB377 | (1:500) |
| Sequence-based reagent | ATF3-F | This paper | PCR primers | CGCCATCCAGAATAAACACC |
| Sequence-based reagent | ATF3-R | This paper | PCR primers | CCTTCAGCTCAGCATTCACA |
| Sequence-based reagent | BDNF-F | This paper | PCR primers | GGTCACAGCGGCAGATAAA |
| Sequence-based reagent | BDNF-R | This paper | PCR primers | TGGGATTACACTTGGTCTCGT |
| Sequence-based reagent | CD3-F | This paper | PCR primers | CCAGTCAAGAGCTTCAGACAA |
| Sequence-based reagent | CD3-R | This paper | PCR primers | GAGTCCTGCTGAGTTCACTTC |
| Sequence-based reagent | CX3CR1-F | This paper | PCR primers | AGTTCCCTTCCCATCTGCTC |
| Sequence-based reagent | CX3CR1-R | This paper | PCR primers | CACAATGTCGCCCAAATAAC |
| Sequence-based reagent | CXCL10-F | This paper | PCR primers | AGCCAAAAAAGGTCTAAAAGGG |
| Sequence-based reagent | CXCL10-R | This paper | PCR primers | CTAGCCATCCACTGGGTAAAG |
| Sequence-based reagent | DCX-F | This paper | PCR primers | GTCACCTGTCTCCATGATTTC |
| Sequence-based reagent | DCX-R | This paper | PCR primers | GACTCTGCATTCATTCTCATCC |

*Continued on next page*

*Continued*

| Reagent type (species) or resource | Designation | Source or reference | Identifiers | Additional information |
|---|---|---|---|---|
| Sequence-based reagent | GDNF-F | This paper | PCR primers | GTGACTCCAATATGCCTGAAGA |
| Sequence-based reagent | GDNF-R | This paper | PCR primers | GCCGCTTGTTTATCTGGTGA |
| Sequence-based reagent | IFNβ-F | This paper | PCR primers | GTGGGAGATGTCCTCAACTG |
| Sequence-based reagent | IFNβ-R | This paper | PCR primers | AGGCGTAGCTGTTGTACTTC |
| Sequence-based reagent | IFNγ-F | This paper | PCR primers | GTTTGAGGTCAACAACCCACAG |
| Sequence-based reagent | IFNγ-R | This paper | PCR primers | GCTTCCTGAGGCTGGATTC |
| Sequence-based reagent | IL-1β-F | This paper | PCR primers | TCGCTCAGGGTCACAAGAAA |
| Sequence-based reagent | IL-1β-R | This paper | PCR primers | CATCAGAGGCAAGGAGGAAAAC |
| Sequence-based reagent | IL-17-F | This paper | PCR primers | GACTACCTCAACCGTTCCAC |
| Sequence-based reagent | IL-17-R | This paper | PCR primers | CACTGAGCTTCCCAGATCAC |
| Sequence-based reagent | FoxP3-F | This paper | PCR primers | GTTCCTTCCCAGAGTTCTTCC |
| Sequence-based reagent | FoxP3-R | This paper | PCR primers | CATCGGATAAGGGTGGCATAG |
| Sequence-based reagent | MIP-1α -F | This paper | PCR primers | AGATTCCACGCCAATTCATC |
| Sequence-based reagent | MIP-1α -R | This paper | PCR primers | CTCAAGCCCCTGCTCTACAC |
| Sequence-based reagent | NGF-F | This paper | PCR primers | ATGCTGGACCCAAGCTCAC |
| Sequence-based reagent | NGF-R | This paper | PCR primers | CTGCCTGTACGCCGATCAAA |
| Sequence-based reagent | NT3-F | This paper | PCR primers | TCACCACGGAGGAAACGCTA |
| Sequence-based reagent | NT3-R | This paper | PCR primers | GTCACCCACAGGCTCTCACT |
| Sequence-based reagent | RORγ -F | This paper | PCR primers | GGATGAGATTGCCCTCTACAC |
| Sequence-based reagent | RORγ -R | This paper | PCR primers | CAGATGTTCCACTCTCCTCTTC |
| Sequence-based reagent | RPL19-F | This paper | PCR primers | GAAGGTCAAAGGGAATGTGTTCA |
| Sequence-based reagent | RPL19-R | This paper | PCR primers | CCTTGTCTGCCTTCAGCTTGT |
| Sequence-based reagent | TNF-α-F | This paper | PCR primers | CTACTGAACTTCGGGGTGATC |
| Sequence-based reagent | TNF-α-R | This paper | PCR primers | TGAGTGTGAGGGTCTGGGC |
| Sequence-based reagent | TRAF3-F | This paper | PCR primers | CAAAGACAAGGTGTTTAAGGATAA |
| Sequence-based reagent | TRAF3-R | This paper | PCR primers | GCCTTCATTCCGACAGTAG |

*Continued on next page*

*Continued*

| Reagent type (species) or resource | Designation | Source or reference | Identifiers | Additional information |
|---|---|---|---|---|
| Sequence-based reagent | Trail -F | This paper | PCR primers | TTTAATTCCAATCTCCAAGGATG |
| Sequence-based reagent | Trail -R | This paper | PCR primers | GATGACCAGCTCTCCATTC |
| Peptide, recombinant protein | MOG$_{35-55}$ | Hooke laboratories | EK-2110 | The peptide used is prepared as emulsion in CFA and provided as reference EK-2110 by Hooke laboratories. |
| Peptide, recombinant protein | TNFα | PeproTech | 315-01A | 250 ng/2 μL |
| Peptide, recombinant protein | IFNγ | PeproTech | 315–05 | 100U/2 μL |
| Commercial assay or kit | GoScript Reverse Transcription kit | Promega | A2791 | |
| Commercial assay or kit | GoTaq qPCR Master Mix | Promega | A6002 | |
| Commercial assay or kit | Bio-Plex Pro(tm) Mouse Cytokine IL-1beta | Biorad | 171G5002M | |
| Commercial assay or kit | Bio-Plex Pro(tm) Mouse Cytokine IL-6 | Biorad | 171G5007M | |
| Commercial assay or kit | Bio-Plex Pro(tm) Mouse Cytokine IL-12p40 | Biorad | 171G5010M | |
| Commercial assay or kit | Bio-Plex Pro(tm) Mouse Cytokine IL-17A | Biorad | 171G5013M | |
| Commercial assay or kit | Bio-Plex Pro(tm) Mouse Cytokine G-CSF | Biorad | 171G5015M | |
| Commercial assay or kit | Bio-Plex Pro(tm) Mouse Cytokine IFNγ | Biorad | 171G5017M | |
| Commercial assay or kit | Bio-Plex Pro(tm) Mouse Cytokine KC | Biorad | 71G5018M | |
| Commercial assay or kit | Bio-Plex Pro(tm) Mouse Cytokine MIP-1alpha | Biorad | 171G5020M | |
| Commercial assay or kit | Bio-Plex Pro(tm) Mouse Cytokine RANTES | Biorad | 171G5022M | |
| Commercial assay or kit | Bio-Plex Pro(tm) Mouse Cytokine TNF-alpha | Biorad | 171G5023M | |
| Other | MOG in CFA emulsion | Hooke laboratories | EK-2110 | The MOG35-55 peptide is prepared as an emulsion in CFA and provided as reference EK-2110 by Hooke laboratories. |
| Other | 'Control' emulsion | Hooke laboratories | CK-2110 | This is the control emulsion without immunizing peptide |
| Other | Fluoromyelin Green | Invitrogen | F34651 | (1:300) |
| Other | Luxol Fast Blue | Sigma | S3382 | |

## Study design

The objective of the present study was to set up an experimental model in mice that was able to recapitulate more closely pathophysiological manifestations encountered in MS, namely a heterogeneity of the immune response and the presence of CLs. Two features absent from the standard EAE model. Secondly, we wanted to create an experimental tool allowing for the study of miRNAs with regards to their potential implication in the formation, extent, and topography of CLs as previous studies in humans put forth a dysregulated miRNA expression in CLs. The study included a series of controlled laboratory experiments carried out in C57BL/6 mice. Several experiments were conducted using a modified version of the EAE model. EAE was induced by immunization with $MOG_{35-55}$ with or without PTX injection. Additionally, in other sets of experiments mice received either vehicle or a mixture of TNFα and INF-γ through stereotactic injection in the somatosensory cortex. A power analysis was used to assess the sample size necessary for the experiments. We took into consideration the known variability of the standard EAE mouse model and the variability expected for the intracortical cytokine injection. Data collection was stopped when an animal reached a predetermined clinical score (see infra) in strict accordance with the European recommendation regarding experimental procedures and the local ethics committee. The data for these mice were excluded from the analysis. Statistical outliers were defined using extreme studentized deviate also known as Grubbs' test. The investigators were not blinded while collecting and analyzing the data except for all the histological analyses that were performed by blinded researchers. The experimental endpoints were determined before the start of said experiment. At the beginning of experiments, mice were weighed and randomly allocated in experimental groups of similar weight. Additional details are provided below.

## EAE induction and clinical score

EAE was induced by immunization with $MOG_{35-55}$ emulsified with CFA using the Hooke Kit (Hooke labs, EK-2110, Lawrence, USA) in 8–10 week-old female C57BL/6 mice (Charles River and Janvier Labs). The emulsion was administered subcutaneously on the upper back, middle back, and lower back (0.1 mL/site). To compare this model to the classical EAE model, in one study, a group of mice received injections of pertussis toxin (PTX, 80 ng/mice) 1 and 2 days post-immunization while pertussis toxin was omitted in the remaining mice of the study. Mice were weighed and scored daily for clinical signs of the disease by a blinded and trained researcher. The scoring was based on the protocol from Hooke laboratories. *Score 0*: No sign of the disease, the tail is erect when picked up at the base, locomotor activity is intact compared to non-immunized mice. *Score 0.5*: when held at the base, the tip of the tail is limp; Score 1: when picked up at the base, the tail is limp; *Score 1.5*: the tail is limp, and walking is slightly wobbly with a weakness in one hind leg. *Score 2*: when holding at the base, the tail is limp and legs are not spread apart or the mouse presents signs of head tilting with poor balance (both symptoms are associated with poor balance). *Score 2.5*: the tail is limp, the two hind legs are weak or no movement in one leg or the mouse presents signs of head tilting with occasional fall over. *Score 3*: the tail is limp with complete or almost complete paralysis of hind legs (legs can 'paddle' but not to move forward of the hip). *Score 3.5*: the tail is limp with complete paralysis of hind legs (hindquarters are flat and the mouse is unable to right itself when put on the side). *Score 4*: the tail is limp with complete paralysis of hind legs associated with weakness or partial paralysis in front legs. *Score 4.5*: limp tail with complete paralysis of hind legs and partial paralysis of the front legs. No movement in the cage, mouse poorly alert. At this stage or for more severe symptoms euthanasia is performed.

In one of the studies, blood was recovered from the submandibular vein at 7- and 14 days post-immunization and once again before euthanasia. Upon euthanasia, mice were perfused with phosphate-buffered saline (PBS, pH 7.4) before tissue harvesting. Experimental procedures were in strict accordance with the European recommendation (2010/63/UE), which was transformed into the Belgian Law of May 29, 2013, regarding the protection of animals used for scientific purposes. The local ethics committee approved the protocol of the study (study agreement 2010/UCL/MD/022, laboratory agreement LA1230314 and study agreement 2017/UCL/MD/024, laboratory agreement LA1230635).

## Intracerebral stereotactic injection

Mice were anesthetized using isoflurane and placed on a stereotactic device. The stereotactic surgery was used to target the primary somatosensory cortex shoulder-neck region (S1ShNc). The skull was exposed through a midline incision and a drill was used to gain access to the brain. The following coordinates were used: bregma + 1 mm caudal, 2 mm laterally from the median sagittal suture, and 1.06 deep in the cortex. A syringe was inserted into the brain and its content was gradually released over a period of 5 min. Once the injection completed, the capillary was slowly removed. Mice received a single intracerebral (i.c.) injection of either 2 µL of a cytokine mixture composed of 250 ng tumor necrosis factor-alpha (TNFα; PeproTech, London, UK), and 100 U of interferon-gamma (IFN-γ; PeproTech, London, UK) dissolved in sterile PBS or vehicle alone. Mice were monitored daily after surgery until euthanasia three days after cytokine injection. Finally, non-immunized mice also received a unilateral stereotactic injection of PBS.

## miRNA profiling

On the day of euthanasia, a subset of mice was perfused with 10 mL RNAlater solution (Life Technologies). The site of injection was micro-dissected out and immediately utilized for RNA extraction. Total RNA was extracted using *miRvana* Kit (Ambion), according to manufacturer protocol. Briefly, tissues were collected in a Cell Disruption Buffer and homogenized with a motorized rotor-stator homogenizer. Subsequently, an organic extraction followed by immobilization of RNA on glass-fiber filters was performed. RNA concentration was measured by NanoDrop UV/VIS microspectrophotometry (ND-1000; NanoDrop Technologies, Wilmington, DE, USA).

Analysis of miRNA expression was performed on a pool of total RNA samples from a subset of mice. Starting from 500 ng of pooled total RNA, a TaqMan MicroRNA Reverse Transcription kit (Applied Biosystems) with Megaplex Pool RT primers (Applied Biosystems) was used. A set of predefined pools of up to 380 stem-looped reverse-transcription (RT) primers enabled the simultaneous synthesis of cDNA from mature miRNAs. The expression profile of miRNAs was performed with TaqMan Microfluidic Array Cards Type A (Applied Biosystems) containing dried TaqMan primers and probes. The experiments were performed on a Viia7 Thermal Cycler (Applied Biosystems). Data were analyzed using Viia7 system software and QuantStudio 3D AnalysisSuite Software (Version 3.1.).

For each miRNA, the amplification curve was checked. The Ct cutoff value of 32 was applied to all miRNA hence miRNA which Ct was above 32 was ignored for the analysis as recommended by the manufacturer. Relative quantification was determined using the ΔΔCt method with U6 as the internal reference. Differential levels of each miRNA were expressed as fold change compared to the control group (when the miRNA was expressed in the CTL group). We also compared miRNA variations between HIS-C and LIS-C groups. Considering specifically miRNA that were increased or decreased at least twofold between HIS-C and LIS-C groups, we performed a pathway analysis of the top enriched canonical pathways associated with the target genes related to the miRNAs dysregulated. These pathways were found by performing the Kyoto Encyclopedia of Genes and Genomes analysis by miRNet, mirPath, or Cytoscape String app databases.

Selected miRNAs were also analyzed in individual mice with RT-qPCR. Briefly, 10 ng of total RNA was retrotranscribed with TaqMan MicroRNA Reverse Transcription Kit (Applied Biosystems) and later amplified with the relevant TaqMan MicroRNA Assay (Applied Biosystems). Experiments were performed using ViiA 7 Real-Time PCR System (Applied Biosystems) and quantification was performed using the ΔΔCt method with U6 as reference. Data were analyzed by Viia7 system software and QuantStudio 3D AnalysisSuite Software (Version 3.1.).

## RT-qPCR

Total RNA from tissues was obtained using TriPure reagent (Roche) according to the manufacturer's instructions. cDNA was synthesized using a reverse transcription kit (Promega, GoScript Reverse Transcription System). Quantitative PCR was performed with a STEPone PLUS Real-Time PCR System (Applied Biosystems) using the SYBR Green mix (Promega, GoTaq qPCR Master Mix) as previously described (*Alhouayek et al., 2013*). Each sample was measured in duplicate during the same run. Products were analyzed by developing a melting curve at the end of the PCR. Data are normalized

to the 60S ribosomal protein L19 (*Rpl19*) used as a reference gene. *Rpl19* mRNA expression was not affected by any of the conditions. The sequences of the primers used are listed in *Table 1*.

## Inflammatory plasma cytokine and chemokine quantification

Blood was harvested from mice at three different time-points (days 7, 14, and 21) post-MOG$_{35-55}$/CFA injection. Plasma cytokines and chemokines IL-1β, IL-6, IL-12(p40), IL-17, TNFα, IFNγ, MIP-1α, RANTES, KC, and G-CSF were quantified using a Bio-Plex Multiplex kit (Bio-Rad, Nazareth, Belgium) and measured by using Luminex technology (Bio-Plex 200; Bio-Rad) following the manufacturer's instructions (*Guillemot-Legris et al., 2016*).

## Histology

Brains and spinal cord were recovered and fixed in 4% PFA for 72 hr, transferred in sucrose solutions (15% then 30%) for cryoprotection, and finally kept at −80°C. Cryosections were cut with a cryostat. The whole brain was serially sliced (sagittal sections) at a thickness of 20 μm and 30 μm (4 slides of 20 μm and 4 slides of 30 μm). Therefore, two successive slices of the brain on the same slide are systematically separated by 180 μm. Each slide analyzed contains 4–5 brain slices (covering a region of 720 to 900 μm (for 4 and 5 brain slices respectively)) and centered on the injection site (PBS or cytokines). A similar strategy was used for the spinal cord. Coronal sections were serially sliced at a thickness of 20 μm and 30 μm (4 slides of 20 μm and 4 slides of 30 μm). Again, two successive slices of the spinal cord on the same slide are systematically separated by 180 μm. Each slide analyzed contains 5–6 spinal cord slices (covering a region of 900 to 1080 μm (for 5 and 6 spinal cord slices respectively)). One slide (brain or spinal cord) of adequate thickness was randomly selected to perform staining. The number of animals used is specified in the figure legends.

## Brightfield staining

Luxol Fast Blue (LFB) stain with Cresyl Violet (CV) counterstain was used for CLs assessment and identification of basic neuronal structure (3-5). LFB staining alone was also used to assess CLs.

**Table 1.** Primer sequences.

| Gene | Product | Forward primer (5' to 3') | Reverse primer (5' to 3') |
|------|---------|---------------------------|---------------------------|
| *Atf3* | ATF3 | CGCCATCCAGAATAAACACC | CCTTCAGCTCAGCATTCACA |
| *Bdnf* | BDNF | GGTCACAGCGGCAGATAAA | TGGGATTACACTTGGTCTCGT |
| *Cd3g* | CD3 | CCAGTCAAGAGCTTCAGACAA | GAGTCCTGCTGAGTTCACTTC |
| *Cx3cr1* | CX3CR1 | AGTTCCCTTCCCATCTGCTC | CACAATGTCGCCCAAATAAC |
| *Cxcl10* | CXCL10 | AGCCAAAAAAGGTCTAAAAGGG | CTAGCCATCCACTGGGTAAAG |
| *Dcx* | DCX | GTCACCTGTCTCCATGATTTC | GACTCTGCATTCATTCTCATCC |
| *Gdnf* | GDNF | GTGACTCCAATATGCCTGAAGA | GCCGCTTGTTTATCTGGTGA |
| *Ifnb1* | IFNβ | GTGGGAGATGTCCTCAACTG | AGGCGTAGCTGTTGTACTTC |
| *Ifng* | IFNγ | GTTTGAGGTCAACAACCCACAG | GCTTCCTGAGGCTGGATTC |
| *Il1b* | IL-1β | TCGCTCAGGGTCACAAGAAA | CATCAGAGGCAAGGAGGAAAAC |
| *Il17a* | IL-17 | GACTACCTCAACCGTTCCAC | CACTGAGCTTCCCAGATCAC |
| *Foxp3* | FoxP3 | GTTCCTTCCCAGAGTTCTTCC | CATCGGATAAGGGTGGCATAG |
| *Ccl3* | MIP-1α | AGATTCCACGCCAATTCATC | CTCAAGCCCCTGCTCTACAC |
| *Ngf* | NGF | ATGCTGGACCCAAGCTCAC | CTGCCTGTACGCCGATCAAA |
| *Ntf3* | NT3 | TCACCACGGAGGAAACGCTA | GTCACCCACAGGCTCTCACT |
| *Rorc* | RORγ | GGATGAGATTGCCCTCTACAC | CAGATGTTCCACTCTCCTCTTC |
| *Rpl19* | RPL19 | GAAGGTCAAAGGGAATGTGTTCA | CCTTGTCTGCCTTCAGCTTGT |
| *Tnf* | TNF-α | CTACTGAACTTCGGGGTGATC | TGAGTGTGAGGGTCTGGGC |
| *Traf3* | TRAF3 | CAAAGACAAGGTGTTTAAGGATAA | GCCTTCATTCCGACAGTAG |
| *Tnfsf10* | Trail | TTTAATTCCAATCTCCAAGGATG | GATGACCAGCTCTCCATTC |

Toluidine blue was used to measure neuronal loss in the cortex. The size of the nucleus was the morphological criteria used to distinguish neurons from glial cells: nuclei area >50 μm$^2$ (neurons). Images were acquired using a Leica DM6000-B microscope or a Leica SCN400 slide scanner.

## Immunofluorescence

Cryosections were washed in PBS containing 0.1% triton (PBS-T), and incubated overnight at 4°C with primary antibodies: anti-MBP (1:1000, Abcam Ab123499), anti-FOXO3 (1:300, Abcam Ab177487), anti-C1q (1:100, Abcam Ab71940), anti-GFAP (1:1000, Dako Z0334), anti-Iba-1 (1:500, Wako 019–19741), anti-CD3 (1:300 Dako A0452), and anti-NeuN (1:500, Millipore MAB377). All primary antibodies were diluted in PBS-T containing 5% normal goat serum (NGS) overnight at 4°C. The following day, slides were washed 3 times for 10 min in PBS and then incubated with the appropriate fluorescent secondary antibody: Alexa-Fluor 488 conjugated goat anti-mouse polyclonal (1:1000, Abcam) and/or Alexa-Fluor 568 conjugated goat anti-rat polyclonal (1:750, Abcam) and/or Alexa-Fluor Plus 647 conjugated goat anti-rabbit polyclonal (1:750, Abcam) in PBS-T containing 5%-NGS overnight at 4°C. Slides were then washed three times for 10 min with PBS, incubated with a nuclear stain (DAPI), and mounted using Vectashield or Fluoromount as mounting medium. Fluoro-myelin staining was used to assess lesions in the spinal cord and cortex. After rehydration in PBS, sections were incubated in Fluoromyelin Green (1:300, Invitrogen F34651) following manufacturer's instructions. Image acquisition was performed using a Pannoramic P250 Flash III slide scanner (3DHISTECH) or a Leica SP8 confocal system with a Z-step increment set to 3 μm. The confocal images presented are the result of the maximum projection of the entire stack.

## Quantification

Image analysis was performed by a blinded researcher using ImageJ software (https://imagej.nih.gov/ij/) (*Guillemot-Legris et al., 2016*). For the area occupied or for the negative area (depending on the output measured), the color image of interest was changed to an 8-bit image. An automatic thresholding method was applied and the value of the area occupied or negative area was obtained. For the brain and the spinal cord, several images were obtained (e.g. left and right sides of the spinal cord, specific regions of the brain for each slice of tissue present on the slide) using all the slices present on the slide (see above) along with the corresponding quantification values. The mean of these values was then calculated to generate a single value per animal. Concerning the fluorescence quantification, the corrected total fluorescence was calculated as follows: integrated density of the area of interest – (size of the area of interest x mean fluorescence of background). Finally, brain fluoromyelin and MBP intensity were normalized further using the corrected total fluorescence of the same region in CTL mice that was set at 100%. For brain analyses, unless otherwise specified, the whole cortical region was considered.

## Statistical analysis

Data distribution was assessed using the omnibus K2 D'Agostino-Pearson normality test. Data were analyzed using Student's unpaired t-test or the Mann-Whitney U-test depending on parametric or non-parametric distribution, and ANOVA (one-way and two-way ANOVA). Compared groups were deemed significantly different if p-value≤0.05. For details please refer to figure legends. Statistical analysis was performed using GraphPad Prism eight software (GraphPad Software, San Diego, USA).

## Acknowledgements

GGM acknowledges the Fondation Charcot (Belgium) for financial support. NSO wishes to thank Prof J Svaren and MJ Rigby (Waisman Center-University of Wisconsin) for their help with the grammar and syntax of the manuscript. MA and OGL are postdoctoral researchers from the FRS-FNRS (Fonds de la Recherche Scientifique) Belgium. PB is a research fellow of the 'Fonds pour la Recherche dans l'Industrie et l'Agriculture' (FRIA, Belgium). We wish to acknowledge the help of Prof E Sokal and J Ravau (UCLouvain) for access to the Luminex, as well as Prof A des Rieux and Dr. L D'auria. We also acknowledge B Buisseret, Y Ben Kouidar and A Laghouati for their skillful help.

## Additional information

### Funding

| Funder | Author |
|---|---|
| Fondation Charcot | Giulio G Muccioli |
| Fonds De La Recherche Scientifique - FNRS | Owein Guillemot-Legris<br>Mireille Alhouayek |
| Fonds pour la Formation à la Recherche dans l'Industrie et dans l'Agriculture | Pauline Bottemanne |

The funders had no role in study design, data collection and interpretation, or the decision to submit the work for publication.

### Author contributions

Nicola S Orefice, Owein Guillemot-Legris, Mireille Alhouayek, Conceptualization, Formal analysis, Validation, Investigation, Writing - original draft, Writing - review and editing; Rosanna Capasso, Pauline Bottemanne, Formal analysis, Investigation; Philippe Hantraye, Michele Caraglia, Funding acquisition; Giuseppe Orefice, Resources, Funding acquisition; Giulio G Muccioli, Conceptualization, Resources, Formal analysis, Supervision, Funding acquisition, Validation, Investigation, Writing - original draft, Writing - review and editing

### Author ORCIDs

Nicola S Orefice (ID) https://orcid.org/0000-0001-8135-3737
Mireille Alhouayek (ID) https://orcid.org/0000-0002-9193-0718
Giulio G Muccioli (ID) https://orcid.org/0000-0002-1600-9259

### Ethics

Animal experimentation: Experimental procedures were in strict accordance with the European recommendation (2010/63/UE), which was transformed into the Belgian Law of May 29, 2013 regarding the protection of animals used for scientific purposes. The local ethics committee approved the protocol of the study (study agreement 2010/UCL/MD/022, laboratory agreement LA1230314 and study agreement 2017/UCL/MD/024, laboratory agreement LA1230635). All surgery was performed under anesthesia, and every effort was made to minimize suffering.

### Decision letter and Author response

Decision letter https://doi.org/10.7554/eLife.56916.sa1
Author response https://doi.org/10.7554/eLife.56916.sa2

## Additional files

### Supplementary files

• Source data 1. miRNA expression (raw data) used for *Supplementary files 1*, *2*, *3*.

• Supplementary file 1. Supplementary Table S1 : List of the miRNA expressed in the different conditions studied.

• Supplementary file 2. Supplementary Table S2: Changes in miRNA expression relative to the control.

• Supplementary file 3. Supplementary Table S3: miRNA expression in the HIS group relative to the LIS group.

• Transparent reporting form

## Data availability

All data generated or analysed during this study are included in the manuscript and supporting files. Source data files have been provided for Tables S1-3.

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
