## [Decision Letter]

**Acceptance summary:**

Your work makes a substantial contribution to understand the development and progression of cortical lesions in multiple sclerosis. Future studies addressing the relevance of miRNA in this important aspect of the MS disease are now necessary.

**Decision letter after peer review:**

Thank you for submitting your article "miRNA profile is altered in a modified EAE mouse model of multiple sclerosis featuring cortical lesions" for consideration by *eLife*. Your article has been reviewed by three peer reviewers, including Markus Kipp as the Reviewing Editor and Reviewer #1, and the evaluation has been overseen by a Catherine Dulac as the Senior Editor.

The reviewers have discussed the reviews with one another and the Reviewing Editor has drafted this decision to help you prepare a revised submission.

Summary:

This is an interesting paper showing new findings in a highly relevant area of research. In this study Nicola orifice and co-workers report that upon cortical cytokine injection, MOG-immunized mice develop cortical lesions, and the severeness of the cortical lesions depends on the initial clinical score. Beyond, the authors provide a miRNA analysis of the experimentally induced cortical lesions and try to correlate the regulated miRNAs with histopathological changes.

The observed findings, although highly relevant for the field, are not entirely new and unexpected. As stated in the Discussion section, in a previous study Merkler and colleagues stereotactically targeted the cerebral cortex by injection of pro-inflammatory mediators in animals that were immunized subclinically with myelin oligodendrocyte glycoprotein and observed the formation of inflammatory cortical lesions. Beyond, several works demonstrated that a focal lesion in the mouse brain can recruit peripheral immune cells to the site of tissue injury. Thus, the observation of the authors that TNF/IFN injections into the neocortex induced cortical lesions could be expected. However, the authors expand these findings and study the regulation of mi-RNA expression in their model. One major limitation of the manuscript is, however, that it remains on the descriptive level. While the authors can show a complex (and sometimes hard to follow) regulation of miRNA species, the functional relevance of these miRNAs for the development and progression of cortical lesions remains speculative.

While most of the comments address missing Information, two reviewers raised doubts about the myelination Level, which should be carefully addressed.

Revisions expected in follow-up work:

Specifically, please address the following Points:

1) The statement that cortical lesions can be predictors of long-term disability is not justified by the cited studies. Please adopt.

2) In the subsection “EAE mouse model with both a heterogeneous immune response and cortical lesions”, the authors state that anti-IBA1 stains microglia. This statement should be adopted as IBA1 is as well expressed by monocytes, which are particularly recruited in EAE.

3) I am not sure whether LFB stains should be used to evaluate myelin loss of grey matter areas. Based on the presentation of Figure 1, I assume that the LFB stain was combined with a Cresyl-Violet stain or some similar one. It appears for me that in the cytokine-injection group (see black box in Figure 3A) there is a dramatic loss in nuclear structures which could indicate neuronal cell loss. Thus, the decrease in what the author call LFB stain does not indicate demyelination but neuronal and/or glial damage. This point is critical to the manuscript and should be addressed by a more in-depth histological and immunohistochemical analysis.

I appreciate that, in parallel, the authors observed a decrease in fluoromyelin and other myelin stains. However, this does not demonstrate true demyelination (i.e. loss of myelin but neuronal/axonal preservation). It might well be that there is neuronal or axonal loss which would result as well in a reduction of anti-myelin staining intensities. To exclude neuronal damage, the authors provide NeuN cell numbers and toluidine-stains. However, the presented data are premature. First, neuronal densities should be demonstrated for control groups, LIS-C and HIS-C groups. In Figure 4—figure supplement 2 and Figure 4H, just the results of the LIS-C and HIS-C groups are demonstrated. This critic applies as well to other stains. Second, it is not clear to me, how the data in Figure 4—figure supplement 2 were evaluated. The graph says "neuronal loss". Compared to what? Beyond, where all cells counted or just cells with a neuronal morphology. If the latter is true, what was the morphological criteria to decide whether a stained cell was a neuron or another cell type?

In the ideal case, design-based stereology should be applied as the method of choice to evaluate total neuronal cell numbers in the cortices. At least,

4) The finding of more cortical lesions in the somatosensory cortex compared to the motor or visual cortex is not surprising as the cytokines, if I understood correctly, were injected into the somatosensory cortex. If the data are relevant for the manuscript, they should be better embedded into the Results section, and their rationale explained.

5) In Figure 7B, FOXO3+ cell numbers are shown in two separate graphs (ipsi versus contra). It is not clear to me why this is necessary as it complicates the interpretation of the results. Beyond, different scales are used which should be avoided.

6) What test(s) was applied to measure data distribution?

7) The authors should provide more information on how the miRNA profiling experiments were evaluated.

8) For each experiment the authors should include information regarding the number of included animals, how many animals were excluded for what reason, and how many independent experiments were performed.

9) It should be NeuN instead of Neun throughout the manuscript.

10) In Figure 1 I suggest to add information regarding what was stained in panel D and also in Figure 3. In Figure 1E the labels are very small in size and hard to read.

11) Please clarify in Figure 2 why is only IL-17 expression relative to PTX?

12) Figure 3A, D and G, Figure 4: Scale bars are missing.

---

## [Author Response]

Revisions for this paper:While most of the comments address missing Information, two reviewers raised doubts about the myelination Level, which should be carefully addressed.

We have carefully adapted the manuscript based on the comments raised by the reviewers (see below for the specific comments).

Although we tried to do so in our original version, we further adapted the text in order to mention only what was measured (e.g. fluoromyelin, MBP immunostaining, etc.). While we agree that additional work, typically using electron microscopy, would unequivocally confirm altered myelination, we used in this work the most frequently used techniques to assess the presence of myelin. We would also like to mention that we mostly mention “cortical lesions” in the manuscript. We do not use “demyelination” or “myelin” to describe our results, as we did not assess this element using for instance electron microscopy.

Revisions expected in follow-up work:Specifically, please address the following Points:1) The statement that cortical lesions can be predictors of long-term disability is not justified by the cited studies. Please adopt.

the reviewer(s) is (are) right. We have corrected the text and adapted the references. We apologize for the mistake.

2) In the subsection “EAE mouse model with both a heterogeneous immune response and cortical lesions”, the authors state that anti-IBA1 stains microglia. This statement should be adopted as IBA1 is as well expressed by monocytes, which are particularly recruited in EAE.

We have corrected the manuscript. Thank you for the reminder.

3) I am not sure whether LFB stains should be used to evaluate myelin loss of grey matter areas. Based on the presentation of Figure 1, I assume that the LFB stain was combined with a Cresyl-Violet stain or some similar one. It appears for me that in the cytokine-injection group (see black box in Figure 3A) there is a dramatic loss in nuclear structures which could indicate neuronal cell loss. Thus, the decrease in what the author call LFB stain does not indicate demyelination but neuronal and/or glial damage.

In Figure 1 we did not report LFB stain. Figure 1—figure supplement 1 shows LFB with Cresyl-Violet in the spinal cord and Figure 3A-C shows LFB and Cresyl-Violet in the cortices. Actually, we used for the first study the classically used LFB + Cresyl-Violet staining and arrived at the same conclusion as the reviewer. Therefore, in additional experiments we wanted to examine these aspects and performed NeuN, Iba-1, CD3 and MBP stainings. In a subsequent cohort (see Figure 4), we used LFB only to avoid these confounding factors.

This point is critical to the manuscript and should be addressed by a more in-depth histological and immunohistochemical analysis.I appreciate that, in parallel, the authors observed a decrease in fluoromyelin and other myelin stains. However, this does not demonstrate true demyelination (i.e. loss of myelin but neuronal/axonal preservation). It might well be that there is neuronal or axonal loss which would result as well in a reduction of anti-myelin staining intensities. To exclude neuronal damage, the authors provide NeuN cell numbers and toluidine-stains. However, the presented data are premature. First, neuronal densities should be demonstrated for control groups, LIS-C and HIS-C groups. In Figure 4—figure supplement 2 and Figure 4H, just the results of the LIS-C and HIS-C groups are demonstrated. This critic applies as well to other stains. Second, it is not clear to me, how the data in Figure 4—figure supplement 2 were evaluated. The graph says "neuronal loss". Compared to what? Beyond, where all cells counted or just cells with a neuronal morphology. If the latter is true, what was the morphological criteria to decide whether a stained cell was a neuron or another cell type?

We fully understand this point. As evidenced by the manuscript title, we focused on cortical lesions (a key feature of MS) and we defined some of their major characteristics in this model. We did evaluate cortical lesions through LFB or MBP staining. However, we know that even though these techniques are strong indicators and are extensively used, they cannot fully replace electron microscopy to quantify demyelination. We feel that further in-depth studies of these cortical lesions would be interesting with regards to myelin loss or axonal preservation for instance but are beyond the scope of our manuscript. A follow-up paper could examine these aspects.

Regarding the neuronal loss, we apologize if we were not clear. As we often did in this manuscript, we used several techniques to assess the same output. Here we used NeuN IF and further used Toluidine Blue staining to address the potential neuronal loss. Regarding Toluidine Blue, we measured the cortical area where neuron nuclei were no longer present and expressed this as a percentage: area of “neuronal loss” / total cortical area *100. The cortical area of neuronal loss was assessed using the size exclusion criteria between neurons and glia nuclei. We changed the figure and legend accordingly.

In the ideal case, design-based stereology should be applied as the method of choice to evaluate total neuronal cell numbers in the cortices. At least,

We agree and, as mentioned above, a follow-up paper could examine these aspects. We would like to respectfully mention, however, that we did not analyze a single slice (i.e. an area) for each brain but several slices, therefore sampling a volume of the brain. Indeed the data we report were obtained by analyzing serial slices of the brain (i.e. 4-5 slices per mouse separated by 180µm each). While our approach is much better than the analysis of a single slice per brain, as mentioned, we agree that using design-based stereology would bring useful information in our follow-up studies. Thank you for the suggestion.

4) The finding of more cortical lesions in the somatosensory cortex compared to the motor or visual cortex is not surprising as the cytokines, if I understood correctly, were injected into the somatosensory cortex. If the data are relevant for the manuscript, they should be better embedded into the Results section, and their rationale explained.

We apologize if the rationale for discussing this point was not clear. Indeed, the intracortical injection of cytokines targeted the somatosensory cortex. We performed this analysis because we wondered if CLs could be found away from the injection site and if differences in their distribution could be found between mice from the HIS-C and LIS-C groups. We amended the manuscript accordingly.

5) In Figure 7B, FOXO3+ cell numbers are shown in two separate graphs (ipsi versus contra). It is not clear to me why this is necessary as it complicates the interpretation of the results. Beyond, different scales are used which should be avoided.

We elected to present FOXO3+ data as two separate graphs (ipsi and contra) as the miRNA data in the same figure (Figure 7A) are split between ipsi and contra. We did so in order to simplify the take home message: we focus first on differences between HIS and LIS in the ipsi-lateral cortex and then in the contra-lateral cortex. Thank you for noticing the scale discrepancy, we have modified the scale as suggested.

6) What test(s) was applied to measure data distribution?

In order to assess the normality of data distribution, we used the omnibus K2 D'Agostino-Pearson normality test. We added this point in the Materials and methods, Statistical analysis section.

7) The authors should provide more information on how the miRNA profiling experiments were evaluated.

Thank you for the comment. We have adapted the Materials and methods section in our revised version. For each miRNA, the amplification curve was checked. The Ct cutoff value of 32 was applied to all miRNA hence miRNA which Ct was above 32 was ignored for the analysis as recommended by the manufacturer. To evaluate miRNA expression and to facilitate data interpretation, all miRNA expressed are displayed in Supplementary file 1 (Ct > 32). The ΔΔCt method was used to compare miRNA expression between experimental groups and these data are reported in Supplementary file 2. Because we focused our work on differences between HIS-C and LIS-C groups, we also show data (using the ΔΔCt method) showing miRNA variations between these two groups either in the ipsi- or in the contra-lateral sides in Supplementary file 3. We also display in Figure 6A the miRNA from Supplementary file 3 that were increased or decreased at least twofold between HIS-C and LIS-C groups. Moreover, we performed a pathway analysis of the top enriched canonical pathways associated with the target genes related to the miRNAs dysregulated in HIS-C-ipsi vs. LIS-C-ipsi (presented in Figure 6A) and found either in HIS-C-ipsi or LIS-C-ipsi (Figure 6B). Finally, in order to link miRNA and their target genes, we schematically represented the interactions between the miRNAs that were found in all three databases and controlling the most genes and a panel of the targeted genes found in at least two databases. Thank you for noticing this, we modified the Materials and methods section.

8) For each experiment the authors should include information regarding the number of included animals, how many animals were excluded for what reason, and how many independent experiments were performed.

The number of animals or N is presented in each figure legends. We reproduced our experimental setup in three different animal cohorts in different laboratories with different experimenters, which is rarely done in in vivo studies. We did not perform the exact same experiment twice; the ethics committee would not allow this. However, using our model in the three different experimental setups, we always obtained an autoimmune heterogeneity (HIS and LIS group) and the presence of cortical lesions.

9) It should be NeuN instead of Neun throughout the manuscript.

Thank you for noticing, we modified the manuscript accordingly.

10) In Figure 1 I suggest to add information regarding what was stained in panel D and also in Figure 3. In Figure 1E the labels are very small in size and hard to read.

Regarding Figure 1D, the information is stated in the figure legend: “Representative photomicrographs of CD3 positive cells (lymphocytes) infiltrating the spinal cord ventral white matter. The scale bar represents 50 µm.”. The same is true for the legends of Figure 3A and C. However, as suggested, we added the information in the figure as well. We also modified the size of the labels in Figure 1E in order to facilitate the reading.

11) Please clarify in Figure 2 why is only IL-17 expression relative to PTX?

We expressed the data as relative to the control group (CTL). However, contrary to the other markers analyzed, IL-17 is not expressed in the CTL group as indicated by the ND (not detected) in the graph. Therefore, we could not express the results of the other groups relative to CTL. As IL-17 is expressed in the other three groups, we decided to report the data for IL-17 as relative to the expression in the PTX group because this is the “classical” EAE model. This is now specified in the legend.

12) Figure 3A, D and G, Figure 4: Scale bars are missing.

Thank you for noticing. We added the scale bars for these figures.